# LORD-GoF: A Robust Online Detection Approach for LLM Watermarks in Sparse and Mixed Streams

**Jiade Xu** [1]  **Zhouping Li** [1]

## Abstract

Watermarking is crucial for identifying AI-generated text; however, existing detection methods often focus on offline settings and fail to control the online False Discovery Rate (oFDR) when applied to real-world streams where machine-generated content is sparse and mixed with human writing. To address this issue, in this paper, we propose LORD-GoF, a novel online detection framework that combines a Goodness-of-Fit (GoF) statistic with the Levels based On Recent Discovery (LORD) procedure. We prove that the LORD-GoF approach can rigorously control the oFDR below a user-specified level by dynamically adjusting detection thresholds. Extensive experiments on watermarked text from Qwen-2.5-3B, Sheared-LLaMA-2.7B, and OPT-1.3B using both the Gumbel-Max and Inverse Transform watermarking schemes show that our method maintains statistical power comparable to offline benchmarks while successfully controlling the oFDR under complex, mixed streaming scenarios.

## 1. Introduction

The rapid development of Large Language Models (LLMs) has significantly changed the field of natural language generation. Modern models like GPT-4, Llama, and Claude can now generate text that closely resembles human writing (Zhang et al., 2022; Touvron et al., 2023; Achiam et al., 2023; Bubeck et al., 2023). While this advancement has opened up many useful applications, it has also increased risks, such as the spread of automated disinformation (Weidinger et al., 2021; Pan et al., 2023; Vykopal et al., 2024), academic dishonesty (Kasneci et al., 2023; Meyer et al.,

[1]School of Mathematics and Statistics, Lanzhou University, Lanzhou, China. Correspondence to: Zhouping Li <lizhp@lzu.edu.cn>.

*Proceedings of the 43rd International Conference on Machine Learning*, Seoul, South Korea. PMLR 306, 2026. Copyright 2026 by the author(s).

2023), and malicious use (Zou et al., 2023). As a result, distinguishing between machine-generated and human-written content has become a critical challenge for maintaining digital trust.

In response, *watermarking* has become a statistically sound solution for tracking content origin. By adding unnoticeable signals when generating tokens, model providers let detection work without sharing model weights. Early methods used biased vocabulary distributions (Kirchenbauer et al., 2023), but recent progress has moved to *adaptive* generation tactics, such as multi-objective optimization (Huo et al., 2024) or entropy-based scaling (Liu & Bu, 2024) to keep text quality better.

However, a significant *methodological mismatch* remains on the *detection* side. Most existing detection methods are tested in *offline* settings, usually assuming a **balanced** class distribution (e.g., an equal mix of positive and negative samples). This assumption contrasts sharply with the real-world conditions of deployment, such as API monitoring or social media filtering. In real-world **streaming** situations, texts $D_1, D_2, D_3, \ldots$ arrive sequentially in a stream. For each document $D_t$, we must immediately decide whether it is human-written ($H_{0,t}$) or machine-generated/watermarked ($H_{1,t}$). In such streams the signal of interest ($H_1$) is naturally sparse, while the null hypothesis ($H_0$) in attribution tasks is broad: for a detector designed for a specific scheme (e.g., Gumbel-Max with secret key $\xi$), all other texts—whether human-written, generated by different AI models, or produced by the same model using a different key—are statistically aligned with the null distribution.

Applying standard static thresholding rules (e.g., a fixed significance level $\alpha = 0.05$) in such sparse streams results in a severe breakdown in error control, a classic example of the **Base Rate Fallacy** (Bar-Hillel, 1980; Kahneman & Tversky, 1973). This statistical issue arises when the low frequency of the signal is overlooked, leading to false positives outweighing true detections, even with a seemingly low Type I error rate.

Consider a **toy example**: in a stream of 105 documents, where 100 belong to the null distribution ($H_0$) and only 5 are truly watermarked ($H_1$). A static detector that rejects when-

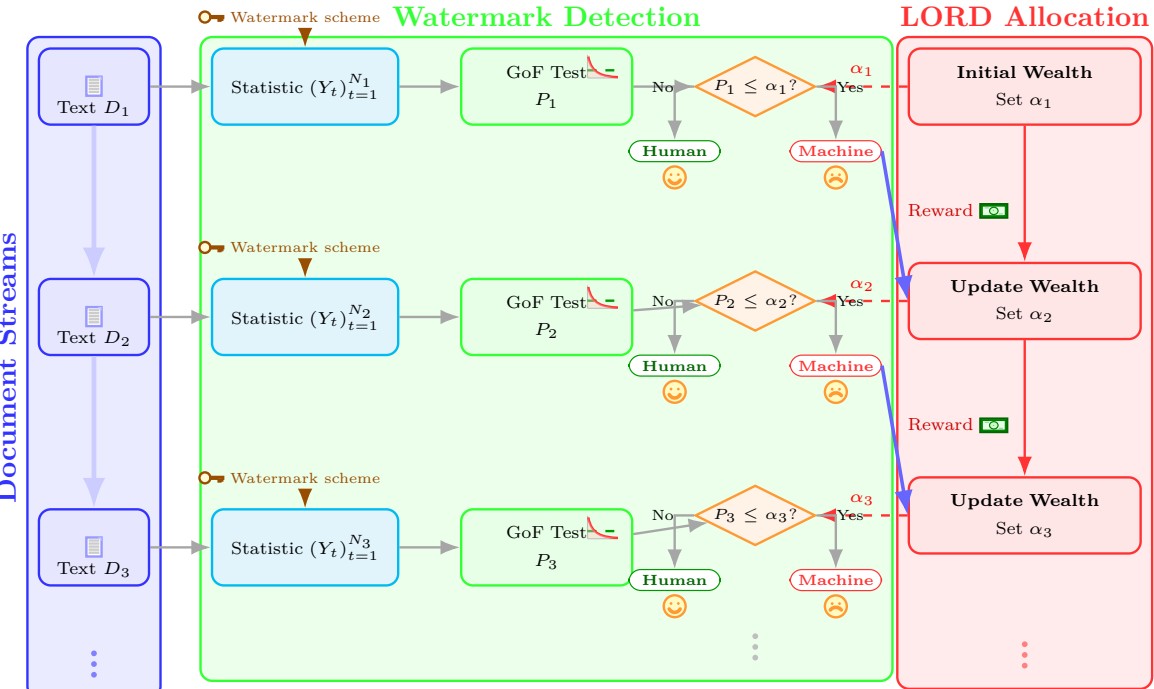

Figure 1. **Schematic of the LORD-GoF Framework.** Left (Blue): Sequentially arriving texts in real-world scenarios, where most are human-written and a small portion are mixed with machine-generated (watermarked) content. Middle (Green): For each incoming document, token-level pivotal statistics are computed to measure whether individual tokens follow the null distribution ($H_0$: human-written). A GoF test aggregates these token-level signals to produce a document-level test statistic quantifying the overall deviation from $H_0$. Right (Red): The LORD procedure dynamically allocates the testing level $\alpha_t$ for each document-level test statistic, ensuring oFDR control while maximizing the detection power for watermarked texts.

ever $P_t \leq \alpha = 0.05$ controls the per-document Type I error rate in expectation, so it flags about $100 \times 0.05 = 5$ null texts as false positives. Even if the detector has perfect power and identifies all 5 watermarked texts, the set of reported discoveries will contain 5 true positives and 5 false positives. The resulting False Discovery Rate is $5/(5 + 5) = 50\%$, an order of magnitude higher than the nominal $5\%$. Offline procedures such as Benjamini–Hochberg do control FDR when all 105 p-values are available simultaneously, but in our streaming setting each decision must be made on arrival, so a fixed-threshold rule does not provide any online FDR guarantee. This example illustrates that offline guarantees are mathematically insufficient for sparse streaming environments, inevitably causing an accumulation of false accusations over time.

To overcome this limitation, we propose LORD-GOF, a unified online detection framework specifically engineered to tackle the aforementioned challenge in streaming environments. As illustrated in Figure 1, the left component of our pipeline corresponds to real-world sequential text arrival, where we must make immediate decisions on each incoming text with no opportunity for subsequent revisions.

The middle component adopts the **Goodness-of-Fit (GoF)** test, proposed in He et al. (2025), which is uniquely suitable for real-world scenarios: an LLM-generated text is likely to undergo human edits (including deletions, additions, and substitutions), and the experiments in He et al. (2025) have demonstrated that GoF tests can achieve robust detection performance in such edited scenarios. The right component employs the **Levels based On Recent Discovery (LORD)** procedure (Javanmard & Montanari, 2018), which dynamically allocates the test level $\alpha_t$. This dynamic allocation ensures that at any time $t$ in the imbalanced data stream, the false discovery rate $FDR_t$ is strictly controlled at $\alpha$, while maximizing the detection power.

**Key Contributions.** Our main contributions are as follows:

- **Novel Online Framework:** We present LORD-GOF in §4, the first framework to combine robust Goodness-of-Fit statistics with online FDR control. We show that this dynamic approach effectively addresses the Base Rate Fallacy, which renders static baselines ineffective in sparse streams.

- **Theoretical Guarantee:** We provide a formal proof

(Theorem 4.1) that LORD-GOF ensures valid online FDR control (oFDR $\leq \alpha$) for any finite time horizon under necessary conditions.

- **Empirical Robustness:** In §5, we demonstrate through extensive experiments on Qwen-2.5-3B (Qwen et al., 2025), Sheared-LLaMA-2.7B (Xia et al., 2024), and OPT-1.3B (Zhang et al., 2022) that LORD-GOF substantially improves false discovery control across various conditions—including different temperatures, sparsity levels, and adversarial attacks—keeping the FDR near or below the target and yielding FDR-valid, high-power statistics in each setting, whereas traditional static methods suffer from severe FDR inflation.

## 2. Related Work

### 2.1. Watermarking for Large Language Models

Text watermarking algorithms typically intervene in the LLM's decoding process to embed statistical signals. Early influential approaches, such as the **Green-Red list (KGW)** scheme proposed by Kirchenbauer et al. (2023), partition the vocabulary to bias the sampling distribution. While effective, such biased sampling can compromise generation quality in low-entropy contexts. Consequently, *distortion-free* or quality-preserving schemes have gained prominence. Key methods include **Gumbel-Max** watermarking (Aaronson, 2023) and the **Inverse Transform** method (Kuditipudi et al., 2023), which preserve the original text distribution while embedding detectable correlations. Notably, Dathathri et al. (2024) introduced **SynthID**, a scalable framework deployed by Google DeepMind that employs a tournament-based sampling strategy.

To further address the trade-off between detectability and text quality, recent research has shifted towards *adaptive* watermarking. For instance, Huo et al. (2024) proposed a token-specific watermarking scheme using lightweight networks and multi-objective optimization to balance semantic coherence and detection z-scores. Similarly, Liu & Bu (2024) introduced an entropy-based adaptive strategy that selectively embeds signals only in high-entropy positions to preserve text quality. These methods highlight the importance of dynamic mechanisms.

Beyond generation, robustness remains a persistent challenge. Qu et al. (2025) introduced provably robust multi-bit watermarks, while Feng et al. (2025) proposed Bi-Mark to enhance detection under various attacks. Furthermore, recent studies have systematically evaluated detection mechanisms, with He et al. (2025) demonstrating that classical **Goodness-of-Fit (GoF)** tests (e.g., Anderson-Darling (Anderson & Darling, 1952), Kuiper (Kuiper, 1960), Kolmogorov-Smirnov (Massey, 1951), Cramér-von Mises (Cramér, 1928), Watson (Watson, 1961), Ney-

man (Neyman, 1937)) significantly outperform standard mean-based Z-tests in capturing distributional anomalies. Additional works explore statistical frameworks (Li et al., 2024), robustness under human edits (Li et al., 2025), watermark stealing (Jovanović et al., 2024), and comprehensive robustness evaluations (Liang et al., 2025).

### 2.2. Online Multiple Hypothesis Testing

The challenge of controlling error rates in sequential decision-making is addressed by the field of Online Multiple Hypothesis Testing. Originating from the seminal *alpha-investing* rule by Foster & Stine (2008) and its generalizations (GAI) (Aharoni & Rosset, 2013), these frameworks manage a dynamic "error budget" (or $\alpha$-wealth). The core principle is that the wealth is consumed by testing and replenished upon rejections, allowing for the discovery of sparse signals in infinite streams.

Javanmard & Montanari (2018) pioneered the **LORD** (Levels based On Recent Discovery) framework, establishing rigorous online FDR control. Building on this, Ramdas et al. (2017) proposed **LORD++**, which uses an asymmetric reward schedule that boosts power after the first discovery. To further improve power when the proportion of non-nulls is non-negligible, adaptive algorithms like **SAF-FRON** (Ramdas et al., 2018) and **ADDIS** (Tian & Ramdas, 2019) were developed. Motivated by the offline Storey-BH procedure (Storey, 2002), these methods incorporate an estimate of the proportion of null hypotheses (using a candidate threshold $\lambda$) to adaptively preserve wealth. Parallel to p-value-based methods, recent works have explored e-values for robust testing under arbitrary dependence (Wang & Ramdas, 2022; Xu & Ramdas, 2024), culminating in the **e-GAI** framework (Zhang et al., 2025). Despite this rich theoretical landscape, these sophisticated online control mechanisms remain significantly underutilized in NLP watermarking, where static thresholds still dominate.

## 3. Preliminaries

In this section, we review the general mechanism of LLM watermarking, GoF Test for Watermark Detection, and the LORD algorithm for online error control.

### 3.1. General Framework of LLM Watermarking

Let $\mathcal{V}$ be the vocabulary. At each generation step $t$, an LLM outputs a next-token prediction (NTP) distribution $\mathcal{P}_t$ over $\mathcal{V}$ conditioned on the preceding context $x_{<t}$. Watermarking embeds cryptographic statistical signals into token generation by inducing a statistical dependency between the sampled token $x_t$ and a pseudo-random seed $\zeta_t$. This seed is generated via a hash function $\mathcal{H}$ with a secret key $\xi$ and contextual tokens: $\zeta_t = \mathcal{H}(x_{t-k:t-1}, \xi)$, where $k$ denotes

*Table 1.* Comparison of three representative watermarking schemes. Symbols: $\mathcal{P}_t$: original next-token prediction (NTP) distribution over vocabulary $\mathcal{V}$; $\zeta_t$: pseudo-random seed (KGW: $G_t$ = Green List subset of $\mathcal{V}$ with $|G_t| = |\mathcal{V}| \cdot \gamma$; Gumbel-Max: $(U^{(w)})_{w \in \mathcal{V}}$ = independent uniform random variables over $(0, 1)$; Inverse Transform: $R_t \sim U(0, 1)$, $\pi_t$ = permutation of $\mathcal{V}$); $H_0$: null hypothesis (human-written text); $H_1$: alternative hypothesis (watermarked text); $F_{\pi_t}(x) = \sum_w \mathcal{P}_t(w) \mathbf{1}\{\pi_t(w) \leq x\}$; $\eta_{\pi_t}(w) = \frac{\pi_t(w)-1}{|\mathcal{V}|-1}$.

| Scheme | Bias Type | $\zeta_t$ | Sampling $x_t = \mathcal{S}(\mathcal{P}_t, \zeta_t)$ | Pivotal Statistic $Y_t$ | Distribution under $H_0$ | Distribution under $H_1$ |
|---|---|---|---|---|---|---|
| **KGW** (Kirchenbauer et al., 2023) | Biased | $G_t$ | $\mathcal{P}' \propto \mathcal{P}_t \cdot e^{\delta \mathbf{1}\{w \in G_t\}}$ | $\mathbf{1}\{x_t \in G_t\}$ | Bernoulli($\gamma$) | Bernoulli($\mu$), $\mu = \frac{\sum_{w \in G_t} \mathcal{P}_t(w) e^\delta}{\sum_{w \in G_t} \mathcal{P}_t(w) e^\delta + \sum_{w \notin G_t} \mathcal{P}_t(w)}$ |
| **Gumbel-Max** (Aaronson, 2023) | Unbiased | $(U^{(w)})_{w \in \mathcal{V}}$ | $\arg\max_w \frac{\log \zeta_t^{(w)}}{\mathcal{P}_t(w)}$ | $\zeta_t^{(x_t)}$ | Uniform$(0, 1)$ | CDF: $\sum_w \mathcal{P}_t(w) r^{1/\mathcal{P}_t(w)}, r \in [0, 1]$ |
| **Inverse Transform** (Kuditipudi et al., 2023) | Unbiased | $(R_t, \pi_t)$ | $\pi_t^{-1}(F_{\pi_t}^{-1}(R_t))$ | $1 - |R_t - \eta_{\pi_t}(x_t)|$ | Triangular$(2y)$, $y \in [0, 1]$ | CDF: $\left(1 - \frac{1-r}{\mathcal{P}_t(1)}\right)^2, r \in [1 - \mathcal{P}_t(1), 1]$ |

the context window size and the seed is only recoverable by authorized verifiers. Instead of sampling $x_t$ directly from $\mathcal{P}_t$, a modified sampler $\mathcal{S}(\mathcal{P}_t, \zeta_t)$ is employed to encode the watermark signal.

For verification, a pivotal statistic $Y_t = Y(x_t, \zeta_t)$ quantifies the dependency between $x_t$ and $\zeta_t$. Valid statistics satisfy that under the null hypothesis ($H_0$: human-written text), $Y_t$ follows a known distribution $\mu_0$ independent of text semantics, while under the alternative ($H_1$: watermarked text), $Y_t$ adheres to an alternative distribution $\mu_{1,\mathcal{P}_t}$ tied to the watermark scheme.

We summarize the core properties of three representative schemes: Green-Red List (KGW), Gumbel-Max (Gum), and Inverse Transform (Inv) in Table 1.

### 3.2. GoF Test for Watermark Detection

When a document $D_t$ of length $N_t$ arrives at time $t$, the verifier tests whether it is consistent with $H_{0,t}$ (human-written) or $H_{1,t}$ (machine-generated/watermarked). A key practical difficulty is that LLM outputs are often post-edited by users—through *substitution, deletion, and insertion*—before they are seen by the verifier (He et al., 2025). Such edits dilute watermark signals because modified tokens no longer follow the watermark-induced randomness, and the verifier typically does not know which tokens were edited (He et al., 2025). This motivates a detection rule whose power is not unduly harmed by mixed human/LLM content and unknown edit patterns.

**Token-level pivotal statistics.** Following the pivotal-statistic framework in He et al. (2025), for each token position $i = 1, \ldots, N_t$, let $w_{t,i}$ be the observed token and let $\zeta_{t,i}$ be the pseudorandomness reconstructed from the text and the secret key. We compute a scheme-specific pivotal statistic

$$Y_{t,i} = Y(w_{t,i}, \zeta_{t,i}) \in \mathbb{R}.$$

Under the null $H_{0,t}$, humans do not have access to $\zeta_{t,i}$, so $w_{t,i}$ is independent of $\zeta_{t,i}$ and $\{Y_{t,i}\}_{i=1}^{N_t}$ are i.i.d. from a known reference distribution with CDF $F_0$ (e.g., Uniform$(0, 1)$ for Gumbel-Max (He et al., 2025)). We stan-

dardize across watermarking schemes via the probability integral transform:

$$u_{t,i} = F_0(Y_{t,i}), \qquad p_{t,i} = 1 - u_{t,i}, \qquad i = 1, \ldots, N_t. \tag{1}$$

Hence, under $H_{0,t}$ we have $u_{t,i} \sim \text{Uniform}(0, 1)$ and therefore $p_{t,i} \sim \text{Uniform}(0, 1)$. Under $H_{1,t}$, watermarking introduces dependence between $w_{t,i}$ and $\zeta_{t,i}$, which shifts the distribution of $Y_{t,i}$ away from $F_0$ and typically makes $p_{t,i}$ concentrate near 0 (He et al., 2025).

**Document-level GoF aggregation.** To aggregate $\{p_{t,i}\}_{i=1}^{N_t}$ into a single document-level score, we apply GoF tests that measure deviation from Uniform$(0, 1)$ under $H_0$. Following (He et al., 2025), we use eight representative tests: **Tr-GoF test (Phi)** (Li et al., 2025), **Kuiper's test (Kui)** (Kuiper, 1960), **Kolmogorov–Smirnov test (Kol)** (Massey, 1951), **Anderson–Darling test (And)** (Anderson & Darling, 1952), **Cramér–von Mises test (Cra)** (Cramér, 1928), **Watson's test (Wat)** (Watson, 1961), **Neyman's smooth test (Ney)** (Neyman, 1937), and **Chi-squared test (Chi)** (Pearson, 1900). Empirically, He et al. (2025) show these GoF tests remain strong under common edits such as word deletion and synonym substitution, and also under stronger "information-rich" edits that selectively modify the most watermarked-looking tokens.

As a representative example, the **Anderson–Darling test (And)** statistic emphasizes tail deviations. Let $p_{(1)} \leq \cdots \leq p_{(N_t)}$ be the sorted p-values. The And statistic is

$$S_t^{\text{And}} = -N_t - \frac{1}{N_t} \sum_{i=1}^{N_t} (2i-1) \left[ \ln p_{(i)} + \ln\left(1 - p_{(N_t+1-i)}\right) \right]. \tag{2}$$

For any chosen GoF statistic $S$, we compute the document-level p-value by calibrating against its null hypotheses under i.i.d. Uniform$(0, 1)$:

$$P_t = \mathbb{P}\left(S \geq S_t^{\text{obs}} \mid H_{0,t}\right), \tag{3}$$

where $S_t^{\text{obs}}$ is the observed statistic on $\{p_{t,i}\}_{i=1}^{N_t}$.

## 3.3. Online FDR Control via LORD

In real-world deployment, texts $D_1, D_2, \ldots$ arrive sequentially. For each $D_t$, an immediate, irreversible decision between $H_{0,t}$ (human-written) and $H_{1,t}$ (watermarked) is required. The Online False Discovery Rate (oFDR) ensures control over the expected proportion of false discoveries at any stopping time $T$, which is defined as:

$$\text{oFDR}(T) = \mathbb{E}\left[\frac{V(T)}{R(T) \vee 1}\right] \leq \alpha,$$

where $V(T)$ is the number of false discoveries and $R(T)$ is the total number of rejections up to $T$.

### 3.3.1. LORD

Javanmard & Montanari (2018) proposed generalized $\alpha$-investing rules and a family of **LORD** procedures , which controls oFDR by dynamically setting the test level $\alpha_t$ through an $\alpha$-*wealth* process $W(t)$. The wealth decreases by $\alpha_t$ when a test is not rejected and increases by a fixed reward $\alpha - W_0$ at each rejection. We use a simple form of it:

$$W(t) = W(t-1) - (1 - R_t)\,\alpha_t + R_t\,(\alpha - W_0), \quad (4)$$

with initial wealth $W(0) = W_0 \in (0, \alpha)$, where $R_t \in \{0, 1\}$ indicates a rejection. The level uses only the most recent discovery time $\tau_{\text{last}}$:

$$\alpha_t = \gamma_{t-\tau_{\text{last}}} W(t-1), \quad (5)$$

where $\{\gamma_j\}_{j \geq 1}$ is non-increasing with $\sum_j \gamma_j = 1$, and $\tau_{\text{last}} = 0$ before the first discovery.

**Why LORD.** The last-discovery rule keeps the update simple and memory-light—each level depends only on the current wealth and $\tau_{\text{last}}$—while still guaranteeing oFDR control. This combination of simplicity and provable control makes LORD a robust default for the long, sparse streams that arise in watermark monitoring, and it is the engine we use throughout.

**Alpha death.** In very sparse streams, long runs without rejections drain $W(t)$ and shrink $\alpha_t \to 0$; each rejection restores wealth, and $\{\gamma_j\}$ keeps more budget on recent steps.

**Hyperparameters.** We set $W_0 = 0.2\alpha$ (so $W_0 = 0.01$ at $\alpha = 0.05$) and $\gamma_j \propto j^{-1.2}$.

### 3.3.2. LORD++

Ramdas et al. (2017) proposed **LORD++**, a refinement of LORD that accumulates wealth from *all* past discoveries $\tau_1 < \tau_2 < \cdots$ (not only the most recent) and uses an asymmetric reward: the first rejection earns $\alpha - W_0$ and every later one earns $\alpha$. The test level becomes

$$\alpha_t = \gamma_t W_0 + (\alpha - W_0)\,\gamma_{t-\tau_1} + \alpha \sum_{j \geq 2} \gamma_{t-\tau_j}, \quad (6)$$

where terms with $\tau_j \geq t$ are dropped. Spreading the budget over all discoveries can make LORD++ more powerful after the first rejection, but in our setting this gain over LORD is marginal (Appendix E); LORD-GoF therefore uses the simpler LORD rule by default and treats LORD++ as a comparison.

### 3.3.3. EXTENSIONS

These extensions handle common streaming settings while keeping strong error control.

**SAFFRON (Ramdas et al., 2018)** SAFFRON was designed for streams where $H_1$ is not extremely rare. It uses a candidate threshold $\lambda$ (default $\lambda = 0.5$) to mark "promising" tests and spend most of the budget on them. In this way, SAFFRON saves $\alpha$-wealth when many hypotheses look null, and often gains power when the stream contains a moderate or large fraction of true signals.

**ADDIS (Tian & Ramdas, 2019)** ADDIS further saves budget by discarding hypotheses that look clearly null (e.g., very large $p_t$ close to 1), using a discard cutoff $\tau_{\text{discard}}$ (default $\tau_{\text{discard}} = 0.9$). The testing level is still chosen from past information, and the discard decision mainly affects future accounting, reducing wasted wealth on obvious $H_0$ cases.

## 4. LORD-GoF Framework

We propose LORD-GoF, a unified framework for *streaming* watermark detection. As illustrated in Figure 1, each incoming document is first converted into a calibrated GoF p-value via the pivotal-statistic mechanism in §3.1–§3.2, and then the resulting p-value stream is processed by LORD in §3.3 to make real-time decisions with online error control.

### 4.1. GoF Watermark Detection

Given an incoming document $D_t$ of length $N_t$, we compute token-level pivotal statistics $\{Y_{t,i}\}_{i=1}^{N_t}$ and standardize them using the probability integral transform as in §3.2:

$$u_{t,i} = F_0(Y_{t,i}), \qquad p_{t,i} = 1 - u_{t,i}.$$

Under $H_{0,t}$, $\{p_{t,i}\}$ are i.i.d. Uniform$(0, 1)$ by construction, while under $H_{1,t}$ watermarking typically shifts them toward 0. We then apply a GoF test to $\{p_{t,i}\}_{i=1}^{N_t}$ and compute the document-level p-value $P_t$ by calibrating the chosen GoF statistic under the i.i.d. Uniform$(0, 1)$ null. Because GoF tests only require a known null, they remain effective even when $D_t$ is partially post-edited (substitution/deletion/insertion), which naturally produces a mixture of null-like and watermark-like tokens (He et al., 2025; Li et al., 2025).

---

**Algorithm 1** LORD-GOF: streaming watermark detection

---

1: **Input:** Stream $D_1, D_2, \ldots$; target level $\alpha$; $W_0 \in (0, \alpha)$; $\{\gamma_t\}_{t \geq 1}$ with $\sum_t \gamma_t = 1$.
2: **Initialize:** $W \leftarrow W_0$; $\tau_{\text{last}} \leftarrow 0$.
3: **for** $t = 1, 2, \ldots$ **do**
4:     **GoF p-value:** compute token p-values $p_{t,i} = 1 - F_0(Y_{t,i})$ and calibrate a GoF test to obtain $P_t$ .
5:     **LORD level:** $\alpha_t \leftarrow \gamma_{t-\tau_{\text{last}}} \cdot W$.
6:     **if** $P_t \leq \alpha_t$ **then**
7:         **Reject** $H_{0,t}$ (flag $D_t$ as watermarked); $W \leftarrow W + (\alpha - W_0)$; $\tau_{\text{last}} \leftarrow t$.
8:     **else**
9:         **Accept** $H_{0,t}$; $W \leftarrow W - \alpha_t$.
10:     **end if**
11: **end for**

---

### 4.2. Online Decision via LORD

The stream $\{P_t\}_{t \geq 1}$ induces an online sequence of hypotheses $H_{0,t}$ vs. $H_{1,t}$. At each time $t$, LORD chooses a predictable threshold $\alpha_t$ based only on past rejections and rejects whenever $P_t \leq \alpha_t$. Specifically, LORD sets $\alpha_t$ according to the update rule in Equation (5), where $W_0 \in (0, \alpha)$ is the initial wealth and $\{\gamma_j\}_{j \geq 1}$ is a non-increasing sequence with $\sum_{j=1}^{\infty} \gamma_j = 1$ (Javanmard & Montanari, 2018).

We summarize the complete LORD-GOF pipeline in Algorithm 1.

We next provide a theoretical guarantee for LORD-GOF. Under the Conditions A1–A3 stated in Appendix A, the GoF calibration ensures that $P_t$ is (conditionally) super-uniform under $H_{0,t}$, and the LORD thresholds $\alpha_t$ are predictable since they depend only on past rejections. Therefore, applying the LORD theory yields valid online error control.

**Theorem 4.1** (oFDR Control). *Under the conditions A1–A3 in Appendix A, for any target level $\alpha \in (0, 1)$ and any valid weighting sequence $\{\gamma_t\}$, LORD-GOF (Algorithm 1) ensures that for all $T$,*

$$\text{oFDR}(T) = \mathbb{E}\left[\frac{V(T)}{R(T) \vee 1}\right] \leq \alpha,$$

*where $V(T)$ is the number of false discoveries and $R(T)$ is the total number of rejections up to time $T$.*

The proof of Theorem 4.1 is provided in Appendix A.

## 5. Experiments

### 5.1. Experimental Setup

**Data & models.** We conduct experiments on the C4 (Colossal Clean Crawled Corpus) dataset (Raffel et al., 2020).[1] We evaluate three open-source LLMs: OPT-1.3B (Zhang et al., 2022), Sheared-LLaMA-2.7B (Xia et al., 2024), and Qwen-2.5-3B (Qwen et al., 2025), employing two distortion-free watermarking schemes: **Gumbel-Max** (Aaronson, 2023) and **Inverse Transform** (Kuditipudi et al., 2023). Unless otherwise stated, the document length is fixed at $N = 400$ tokens.

**Streaming protocol and Parameters.** We simulate an online stream of documents $\{D_t\}_{t=1}^M$ over a finite horizon $M$, where the generation dynamics are governed by three pivotal parameters varied across our experiments. We define **Global Sparsity** ($\pi$) as the prior probability that a document contains a watermark (i.e., $\mathbb{P}(H_{1,t}) = \pi$), thereby controlling the rarity of valid signals in the stream. To model the intensity of post-generation attacks or edits, we introduce **Local Density** ($\rho$), which denotes the fraction of tokens in a watermarked document that remain uncorrupted (i.e., the mixture rate). Finally, we vary the decoding **Temperature** ($\tau$) to modulate the generation entropy, which directly influences the statistical strength of the embedded watermark signal. **GoF tests.** For each document, we compute the pivotal statistics and derive a document-level p-value $P_t$ (§3.2). We utilize the suite of eight GoF tests identified in He et al. (2025): **Tr-GoF (Phi)**, **Kuiper (Kui)**, **Kolmogorov–Smirnov (Kol)**, **Anderson–Darling (And)**, **Cramér–von Mises (Cra)**, **Watson (Wat)**, **Neyman (Ney)**, and **Chi-squared (Chi)**.

**Compared methods.** We compare two distinct detection paradigms: (i) **Naive-Fixed**, a static approach that rejects the null hypothesis whenever $P_t \leq \alpha$ (standard $\alpha = 0.05$); and (ii) **LORD-GOF**, our proposed online method which rejects when $P_t \leq \alpha_t$, where $\alpha_t$ is dynamically adjusted by the LORD algorithm (§3.3). For LORD, we use a standard configuration with initial wealth $W_0 = 0.01 \, (= 0.2\alpha)$ and a decay sequence $\gamma_t \propto t^{-1.2}$, unless specified otherwise. Document-level p-values $P_t$ are calibrated against the i.i.d. Uniform$(0, 1)$ null by Monte Carlo with $2 \times 10^4$ resampled null token vectors of length $N_t$. A comparison against alternative online FDR procedures (LORD, SAFFRON, ADDIS, and e-value variants) is reported in Appendix E. We report the empirical False Discovery Rate (FDR) and Power at the end of the stream.

**Additional results.** Detailed ablation studies, runtime analysis, and stability tests across different models and parameters are provided in the Appendix; see the appendix table of

---

[1]Code available at https://github.com/5683259/LORD-GoF-Watermark.

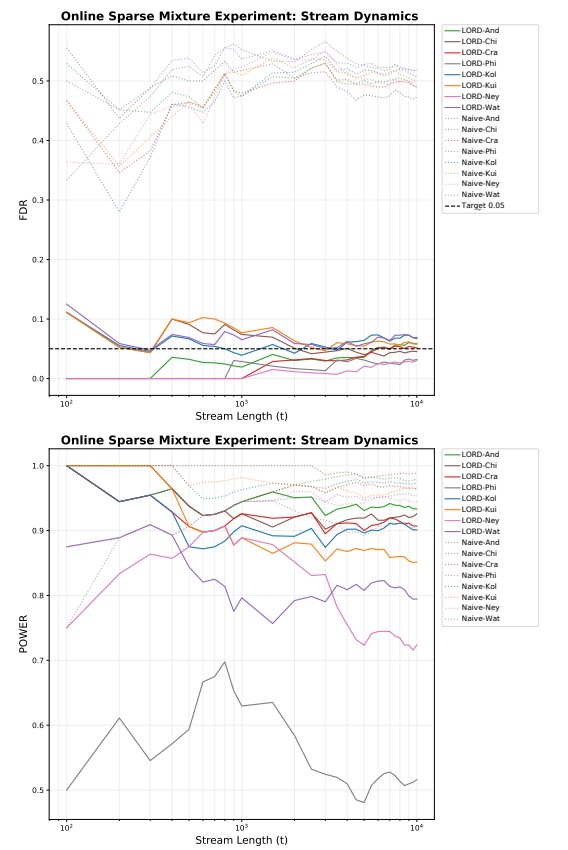

*Figure 2.* **Streaming detection dynamics** (OPT-1.3B, $\tau = 0.5$).

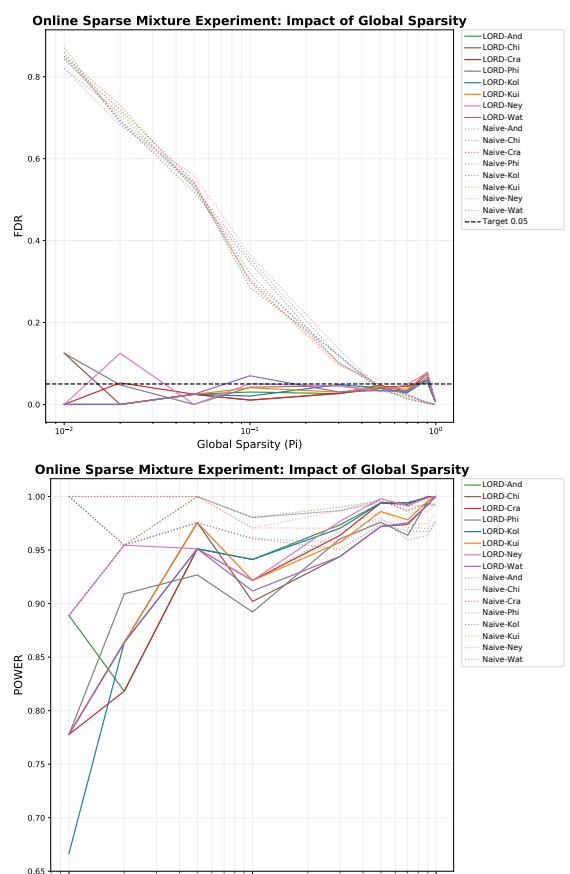

*Figure 3.* **Impact of global sparsity** $\pi$ (Qwen-2.5-3B, $\tau = 0.7$).

contents on its first page for an outline.

## 5.2. Long-term Streaming Dynamics

To evaluate the stability of LORD-GOF over extended periods, we simulate a continuous monitoring scenario with a stream length of $M = 10,000$. **Setup.** We utilize `OPT-1.3B` with **Gumbel-Max** watermarking at a generation temperature of $\tau = 0.5$, representing a standard medium-entropy generation setting. The stream is configured as a *sparse mixture*: the global fraction of watermarked documents is low ($\pi = 0.05$), and watermarked instances are subject to moderate human editing, modeled by a local watermark density of $\rho = 0.7$ (i.e., 30% of tokens are corrupted). This setup mimics a realistic API monitoring task where the vast majority of traffic is human-written ($H_0$).

**Results.** Figure 2 reveals a catastrophic failure of offline detection methods relying on static thresholds. As shown, all Naive-Fixed methods converged to an unacceptable False Discovery Rate (FDR) of approximately 0.50. This occurs because fixed significance levels accumulate Type I errors linearly over time, causing false positives to rival true discoveries in sparse regimes. In stark contrast, the LORD-GOF framework successfully mitigated this error

accumulation. By dynamically penalizing thresholds after non-rejections, it kept the FDR bounded at or near the target $\alpha = 0.05$. Among the evaluated statistics, the **Anderson-Darling (And)** and **Chi-Squared** ($\chi^2$) tests emerged as the most robust pivotal statistics, achieving a superior balance with high sensitivity (Power $> 0.92$) and tight error control, whereas other metrics like Phi proved overly conservative with significantly reduced detection power.

## 5.3. Impact of Global Sparsity ($\pi$)

The global sparsity $\pi$ (the proportion of watermarked traffic) governs the rate at which the online procedure can "replenish" its alpha-wealth. We vary $\pi$ from 0.01 (extremely sparse) to 0.99 (dense) to evaluate robustness.

**Setup.** For this experiment, we utilize `Qwen-2.5-3B` equipped with the **Inverse Transform** watermark. We fix the generation temperature at $\tau = 0.7$ and the local watermark density at $\rho = 0.7$. For each sparsity level $\pi$, we simulate a stream of length $M = 1,000$ to assess the asymptotic behavior of FDR and Power.

**The Sparsity Trap for Static Thresholds.** Figure 3 illus-

*Table 2.* Detection performance for **Gumbel-Max** watermarking on `Sheared-LLaMA-2.7B`. Values are reported as FDR / Power. **Bold** indicates the highest Power among methods with FDR $\leq 0.05$.

| Method | $\tau = 0.1$ | | $\tau = 0.3$ | | $\tau = 0.5$ | | $\tau = 0.7$ | | $\tau = 0.9$ | |
|---|---|---|---|---|---|---|---|---|---|---|
| | FDR | Power | FDR | Power | FDR | Power | FDR | Power | FDR | Power |
| *Offline (Naive-GoF, $\alpha = 0.05$)* | | | | | | | | | | |
| Naive-Kol | 0.593 | 0.796 | 0.506 | 0.786 | 0.532 | 0.902 | 0.430 | 0.984 | 0.570 | 1.000 |
| Naive-Kui | 0.600 | 0.864 | 0.506 | 0.786 | 0.613 | 0.878 | 0.433 | 0.952 | 0.538 | 1.000 |
| Naive-Cra | 0.614 | 0.773 | 0.456 | 0.768 | 0.493 | 0.927 | 0.418 | 0.968 | 0.543 | 1.000 |
| Naive-And | 0.622 | 0.773 | 0.443 | 0.786 | 0.487 | 0.927 | 0.406 | 0.968 | 0.538 | 1.000 |
| Naive-Wat | 0.632 | 0.796 | 0.544 | 0.750 | 0.598 | 0.854 | 0.424 | 0.919 | 0.561 | 1.000 |
| Naive-Chi | 0.526 | 0.841 | 0.494 | 0.804 | 0.542 | 0.927 | 0.456 | 1.000 | 0.506 | 1.000 |
| Naive-Ney | 0.559 | 0.932 | 0.529 | 0.875 | 0.537 | 0.927 | 0.444 | 0.968 | 0.533 | 1.000 |
| Naive-Phi | 0.506 | 0.932 | 0.525 | 0.857 | 0.568 | 0.854 | 0.487 | 0.952 | 0.557 | 1.000 |
| *Online (LORD-GoF, $\alpha = 0.05$)* | | | | | | | | | | |
| LORD-Kol | 0.065 | 0.659 | 0.030 | 0.571 | 0.029 | 0.829 | 0.081 | 0.919 | 0.065 | 1.000 |
| LORD-Kui | 0.030 | 0.727 | 0.057 | 0.589 | 0.000 | 0.707 | 0.052 | 0.887 | **0.023** | **1.000** |
| LORD-Cra | 0.000 | 0.591 | 0.029 | 0.589 | 0.030 | 0.781 | 0.079 | 0.936 | 0.085 | 1.000 |
| LORD-And | 0.000 | 0.591 | 0.029 | 0.589 | **0.000** | **0.854** | 0.077 | 0.968 | 0.104 | 1.000 |
| LORD-Wat | 0.000 | 0.659 | 0.094 | 0.518 | 0.036 | 0.659 | **0.035** | **0.887** | 0.046 | 0.977 |
| LORD-Chi | **0.029** | **0.750** | 0.000 | 0.607 | 0.029 | 0.805 | 0.094 | 0.936 | **0.023** | **1.000** |
| LORD-Ney | 0.068 | 0.932 | 0.060 | 0.839 | 0.081 | 0.829 | 0.053 | 0.871 | 0.046 | 0.977 |
| LORD-Phi | 0.068 | 0.932 | **0.000** | **0.839** | 0.063 | 0.732 | 0.019 | 0.823 | 0.047 | 0.954 |

trates a critical vulnerability in offline detection methods. We observe an inverse relationship between sparsity and false discovery rates for static thresholds. Specifically, in the extremely sparse regime ($\pi = 0.01$), Naive-Fixed methods suffer from a catastrophic FDR inflation, reaching as high as $0.85$ (e.g., *Naive-And* and *Naive-Kol*). This confirms that fixed thresholds are statistically invalid for real-world API monitoring, where watermarked content is expected to be rare.

**Adaptive Control via Wealth Management.** In contrast, LORD-GoF demonstrates exceptional adaptivity. In the sparse regime ($\pi = 0.01$), it successfully throttles the significance thresholds, suppressing the FDR to $0.00$ while maintaining a respectable Power of approximately $0.89$ (using the Anderson-Darling statistic). As the stream becomes denser ($\pi \to 0.99$), the frequent discovery of watermarks allows LORD-GoF to accumulate sufficient wealth, eventually converging to the maximal power ($1.0$) exhibited by the Naive methods. This confirms that LORD-GoF provides safety in sparse regimes without sacrificing performance in dense regimes.

### 5.4. Impact of Generation Temperature ($\tau$)

Temperature $\tau$ controls decoding entropy and significantly affects watermark strength. We evaluate `Sheared-LLaMA-2.7B` with the **Gumbel-Max** water-

mark, varying $\tau \in \{0.1, 0.3, 0.5, 0.7, 0.9\}$ and the local watermark density at $\rho = 0.7$. Table 2 reports the final FDR and Power for each statistic. Note that following the convention in He et al. (2025), we map the Greenwood statistic to **Phi** and Rao's statistic to **Ney** in our results. **Bold** highlights the highest power among methods that successfully control FDR $\leq 0.05$.

**Results Analysis.** Table 2 shows that the best GoF statistic depends strongly on decoding entropy. At low entropy ($\tau = 0.1, 0.3$), spacing/binning-style tests are most reliable: at $\tau = 0.1$, **Chi-squared (Chi)** is the strongest FDR-valid choice (Power 0.750, FDR 0.029), while at $\tau = 0.3$ **Tr-GoF (Phi)** achieves the best performance (Power 0.839, FDR 0.000). At medium entropy ($\tau = 0.5, 0.7$), tail behavior matters more: **Anderson–Darling (And)** is optimal at $\tau = 0.5$ (Power 0.854, FDR 0.000), but at $\tau = 0.7$ several sensitive tests lose FDR control and **Watson (Wat)** becomes the most robust (Power 0.887, FDR 0.035). At high entropy ($\tau = 0.9$), detection becomes easier and both **Kuiper (Kui)** and **Chi-squared (Chi)** reach perfect power (1.000) with low FDR (0.023). Overall, this shifting optimality highlights the value of LORD-GoF, which can pair online control with a flexible library of GoF statistics across generation regimes.

# 6. Conclusion

We introduce LORD-GOF, a practical framework for watermark detection in streaming text. The key idea is to combine robust Goodness-of-Fit (GoF) p-values with online FDR control, so the detector can adapt its threshold over time. This directly fits real deployments, where watermarked content is rare and texts may be mixed with human writing. Compared with static thresholds, LORD-GOF keeps the online false discovery rate under control while maintaining strong detection power, making it suitable for long-running monitoring.

# Acknowledgements

We sincerely thank Xiangkun Wu for his valuable discussions, and the anonymous reviewers for their valuable and insightful comments, all of which have significantly improved the paper.

# Impact Statement

LORD-GOF connects online FDR control with LLM watermark detection, providing a statistically grounded tool for real-time monitoring. It can support content moderation and provenance tracking for platforms that process text streams, while reducing the risk of incorrectly flagging human-written content. More broadly, it offers a foundation for building adaptive and reliable watermark detectors in real-world AI systems.

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

# Appendix Contents

## A. Theoretical Guarantees

In this section, we establish the theoretical guarantee for LORD-GoF. Our analysis integrates the robust watermark detection framework from Li et al. (2025) with the martingale-based online FDR control theory for LORD established by Javanmard & Montanari (2018).

### A.1. Statistical Model and Assumptions

Let $\mathcal{F}_t$ denote the filtration generated by all information available up to time $t$. We verify the validity of our procedure under the following conditions:

- **Condition A1 (Null Hypothesis $H_0$ Perfect Pseudorandomness):** Under the null hypothesis $H_{0,t}$ (human-written text), the watermark keys behave as independent random oracles. Consequently, the pivotal statistics $\{u_{t,i}\}_{i=1}^n$ are independent and identically distributed (i.i.d.) according to $\mathcal{U}[0,1]$.

- **Condition A2 (Alternative Hypothesis $H_1$ Sparse Mixture):** Under the alternative $H_{1,t}$ (watermarked text with edits), the document is modeled as a mixture. The statistics are drawn from $u_{t,i} \sim \varepsilon_n F_W + (1 - \varepsilon_n)\mathcal{U}[0,1]$, where $\varepsilon_n$ is the non-null fraction.

- **Condition A3 (Stream Independence):** The null p-values are mutually independent. Specifically, for any $t$ where $H_{0,t}$ is true, $P_t \perp \mathcal{F}_{t-1}$.

**Relaxing the independence assumption.** Condition A3 is the cleanest sufficient condition for LORD to control oFDR via the GAI martingale argument, but it can be substantially relaxed without altering the LORD-GoF pipeline.

- *Local positive dependence.* Fisher (2024) show that for *LORD++* and SAFFRON, online FDR control extends from independent null p-values to a *local* form of positive dependence. The original LORD rule used here retains the independence/conditional-super-uniformity guarantee; streams that are positively dependent in the stronger PRDS sense are instead covered by LOND (and, offline, by Benjamini–Hochberg), while arbitrary dependence is handled by the e-value route below.

- *Arbitrary dependence via e-values.* Zhang et al. (2025) extend the GAI framework to e-values and obtain online FDR control under *arbitrary* dependence as long as the e-values are conditionally valid. In Appendix E we instantiate this route by feeding our document-level p-values into the standard calibrator $e_t = \kappa\, p_t^{\kappa-1}$ (with $\kappa \in (0,1)$; we use $\kappa = 0.5$, i.e. $e_t = 0.5/\sqrt{p_t}$), which integrates to 1 under a uniform null and is therefore a conditionally valid e-value, and applying e-LORD / e-SAFFRON.

In all reported experiments, the secret-key hashing used by the watermarking schemes ensures that under $H_0$ token-level pivotal statistics are i.i.d. uniform by construction, so document-level p-values inherit exact uniformity. Stream-level independence is therefore the only non-trivial part of Condition A3 in practice.

### A.2. Main Theorem and Proof

**Theorem A.1** (oFDR Control and Consistency)**.** *Suppose Conditions A1–A3 hold. For any target $\alpha \in (0,1)$ and valid weighting sequence $\{\gamma_t\}$, the* LORD-GoF *procedure guarantees:*

$$\mathrm{oFDR}(T) \leq \alpha, \quad \forall T \in \mathbb{N}. \tag{7}$$

*Proof.* The proof instantiates the online-FDR (alpha-investing) argument under which LORD was originally analyzed (Javanmard & Montanari, 2018) for our calibrated GoF p-values.

**Step 1: Super-Uniformity under $H_0$.** Consider any $t$ where $H_{0,t}$ holds. By Condition A1, the token statistics are i.i.d. uniform. Since the document-level p-value $P_t$ is derived via the Probability Integral Transform of a continuous GoF statistic on uniform data, $P_t$ follows $\mathcal{U}[0,1]$ exactly. Combining this with Condition A3 (Independence), we have the conditional super-uniformity property:

$$\mathbb{P}(P_t \leq x \mid \mathcal{F}_{t-1}, H_{0,t}) \leq x, \quad \forall x \in [0,1]. \tag{8}$$

**Step 2: Martingale Control via GAI.** The LORD update rule in Algorithm 1 corresponds to a GAI rule where the wealth invested at step $t$ is $\varphi_t = \alpha_t$. As shown in Javanmard & Montanari (2018), provided that the p-values are conditionally super-uniform (Step 1) and the thresholds $\alpha_t$ are predictable ($\mathcal{F}_{t-1}$-measurable), the total wealth spent is bounded by the initial wealth plus earned wealth. Specifically, let $V(T)$ be the false discoveries. The expectation of the false discovery proportion is bounded by:

$$\mathbb{E}\left[\frac{V(T)}{R(T) \vee 1}\right] \leq \alpha. \tag{9}$$

**Step 3: Asymptotic Consistency.** Steps 1–2 give FDR control under A1–A3 regardless of signal strength. Power requires the watermark to be detectable: if the edit rate scales as $\varepsilon_n \asymp n^{-p}$ and the signal singularity as $\Delta_n \asymp n^{-q}$ with $q + 2p < 1$, then Li et al. (2025) showed the Type I and Type II errors of Tr-GoF tests converge to 0. Thus, even as LORD adjusts $\alpha_t$, for sufficiently large $n$, $P_t < \alpha_t$ holds with probability $\to 1$, ensuring rejections occur and wealth is replenished. $\square$

## B. Additional Simulation Experiments

Here, we extend evaluations to three LLMs (`OPT-1.3B`, `Qwen-2.5-3B`, `Sheared-LLaMA-2.7B`) and two watermark implementations (Gumbel-Max and Inverse Transform), validating the generalizability of our framework across model architectures and watermark designs.

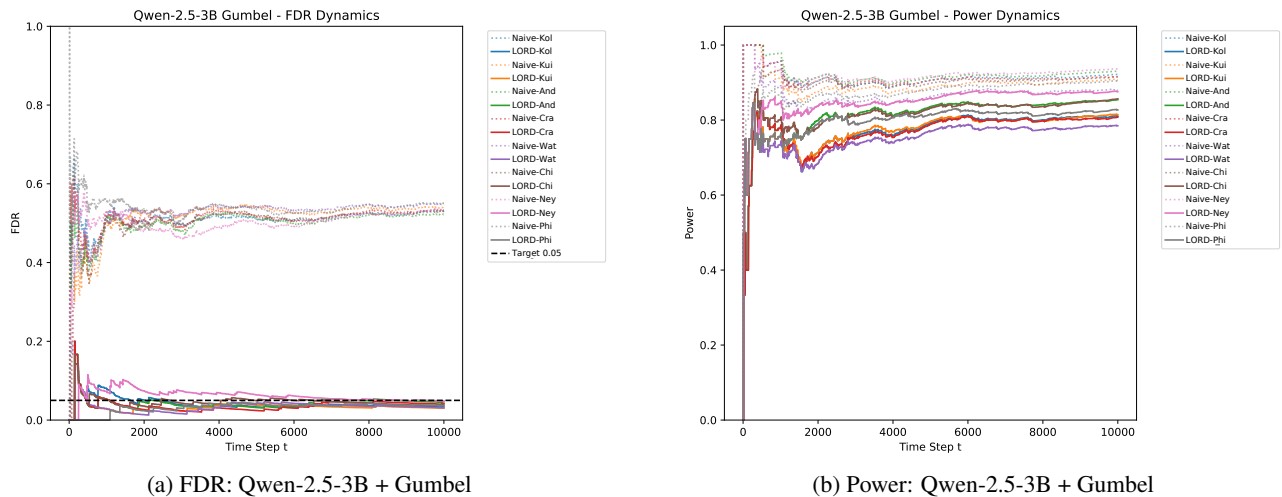

(a) FDR: Qwen-2.5-3B + Gumbel       (b) Power: Qwen-2.5-3B + Gumbel

*Figure 4.* **Dynamics for Qwen-2.5-3B (Gumbel-Max).**

## B.1. Simulation Setup

Default parameters:

- Stream Length: $M = 1,000$ documents

- Temperature: $\tau = 0.7$

- Sequence Length: $N = 400$ tokens/document

- Sparsity: $\pi = 0.05$ (global watermarked document proportion)

- Mixing: $\rho = 0.7$ (local watermarked-token density; the human edit/corruption rate is $r = 1 - \rho$)

We analyze four key experiments (tested across all three models and both watermarks):

1. Long-term stream dynamics ($M = 10,000$, FDR/Power stability over $t = 1 \ldots 10,000$)

2. Global sparsity impact ($\pi \in [0.01, 0.99]$)

3. Temperature impact ($\tau \in \{0.1, 0.3, 0.5, 0.7, 0.9\}$)

4. Local density impact ($\rho \in \{0.8, 0.9\}$)

## B.2. Long-term Stream Dynamics

We present the long-term streaming dynamics for the remaining five model-watermark configurations not covered in the main text in Figures 4–8. For each setting, we simulate a stream of length $M = 10,000$ with fixed sparsity $\pi = 0.05$ and compare the 8 static baselines (**Naive-Fixed**, dotted lines) against our 8 online methods (**LORD-GOF**, solid lines). As shown in these figures, the Naive methods' FDR consistently exceeds 0.6 (gray dotted line), while LORD-GOF keeps the FDR at or close to the target level $\alpha = 0.05$ across all models and watermarks.

## B.3. Impact of Global Sparsity ($\pi$)

We further investigate the robustness of LORD-GOF across a wide range of global sparsity levels $\pi \in [0.01, 0.99]$, with results for the five additional configurations presented in Figures 9–13. These findings reinforce the "sparsity trap" observation: while static methods (dotted lines) consistently fail to control FDR in sparse regimes ($\pi < 0.1$), often reaching error rates near 1.0 and confirming their unsuitability for rare-event detection, LORD-GOF (solid lines) adapts perfectly. It conservatively suppresses rejections to maintain FDR $\leq 0.05$ in sparse streams, yet successfully "recharges" its wealth in denser regimes ($\pi \to 0.99$) to converge to the offline power upper bound.

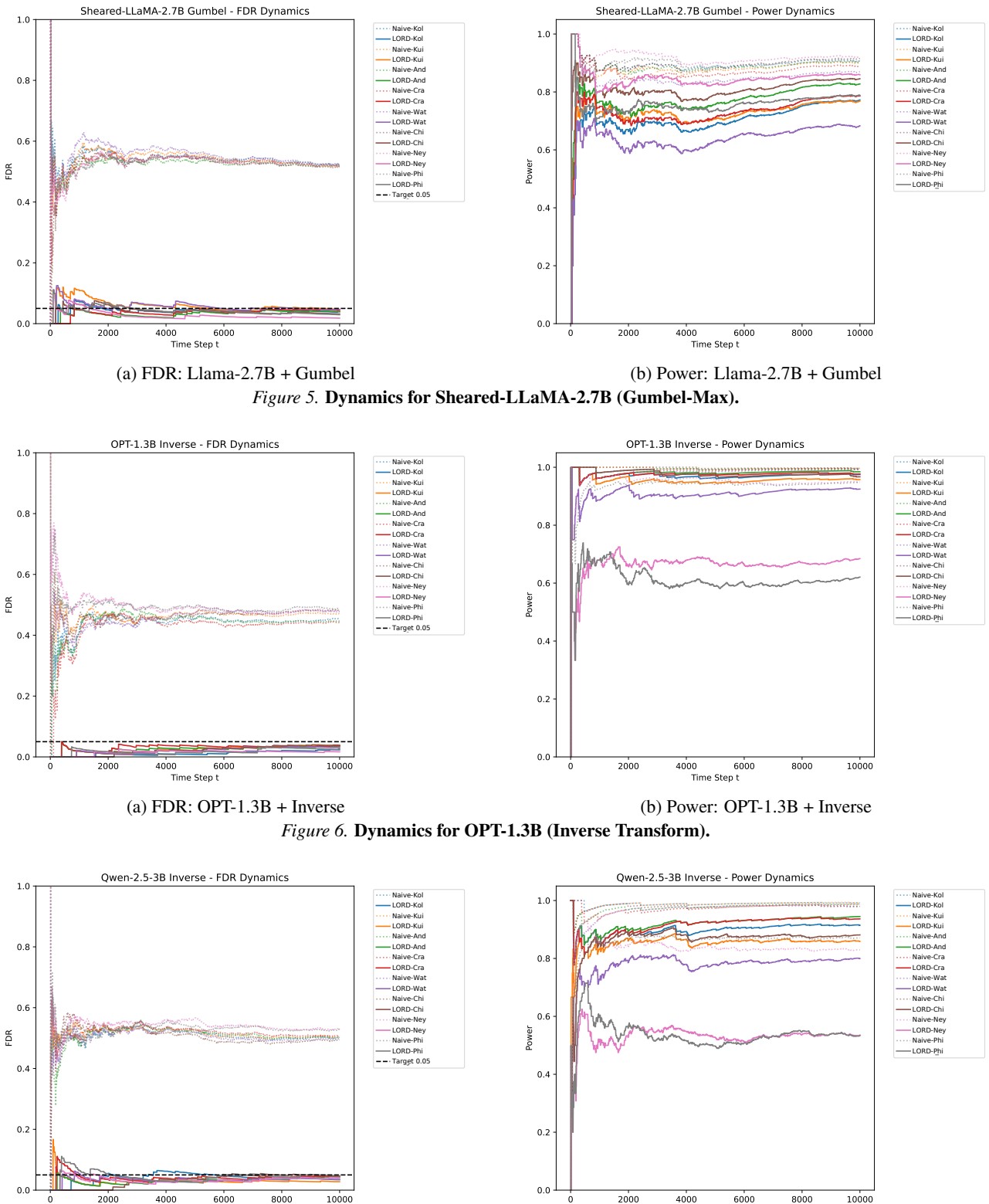

(a) FDR: Llama-2.7B + Gumbel

(b) Power: Llama-2.7B + Gumbel

*Figure 5.* **Dynamics for Sheared-LLaMA-2.7B (Gumbel-Max).**

(a) FDR: OPT-1.3B + Inverse

(b) Power: OPT-1.3B + Inverse

*Figure 6.* **Dynamics for OPT-1.3B (Inverse Transform).**

(a) FDR: Qwen-2.5-3B + Inverse

(b) Power: Qwen-2.5-3B + Inverse

*Figure 7.* **Dynamics for Qwen-2.5-3B (Inverse Transform).**

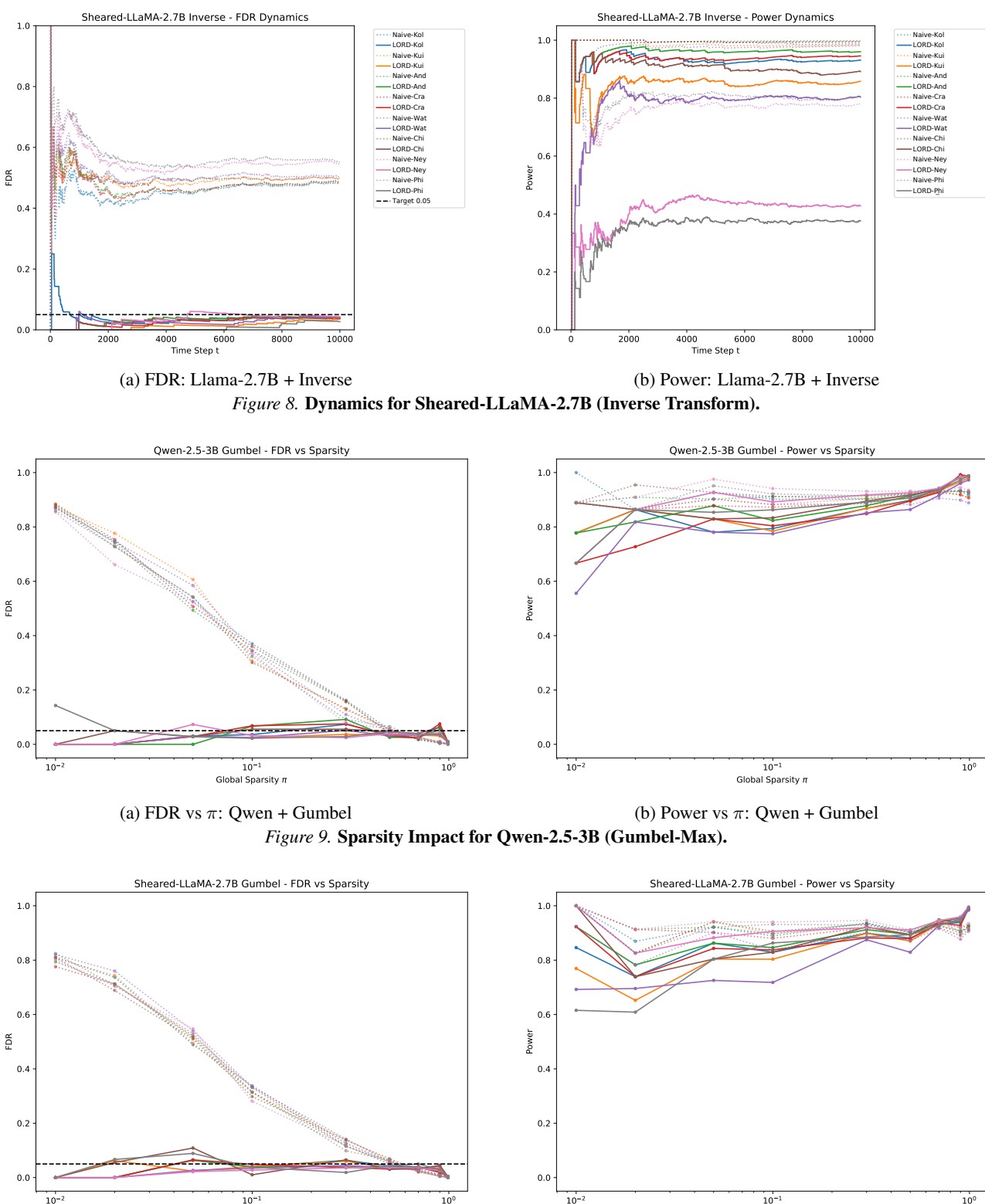

(a) FDR: Llama-2.7B + Inverse

(b) Power: Llama-2.7B + Inverse

*Figure 8.* **Dynamics for Sheared-LLaMA-2.7B (Inverse Transform).**

(a) FDR vs $\pi$: Qwen + Gumbel

(b) Power vs $\pi$: Qwen + Gumbel

*Figure 9.* **Sparsity Impact for Qwen-2.5-3B (Gumbel-Max).**

(a) FDR vs $\pi$: Llama + Gumbel

(b) Power vs $\pi$: Llama + Gumbel

*Figure 10.* **Sparsity Impact for Sheared-LLaMA-2.7B (Gumbel-Max).**

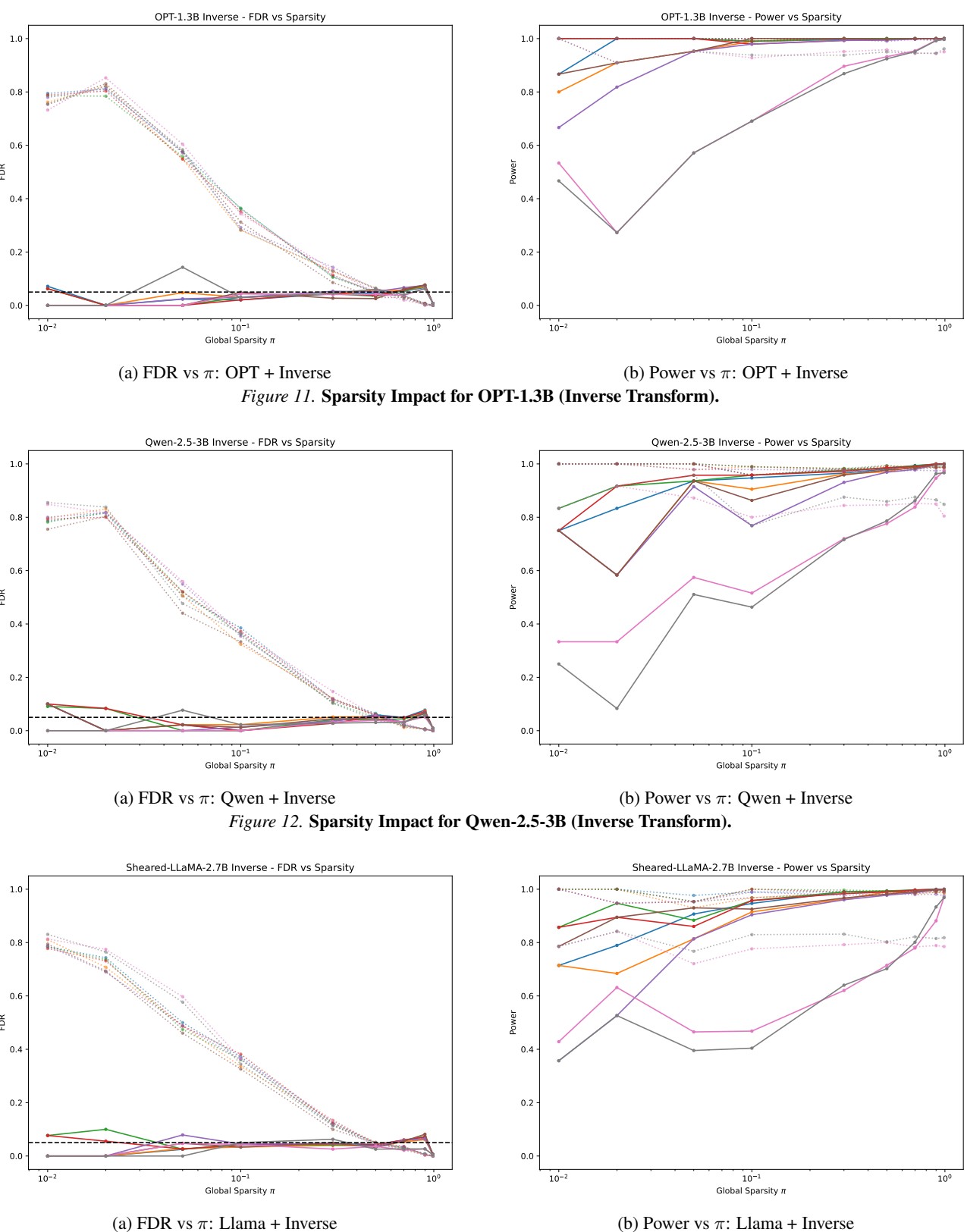

(a) FDR vs π: OPT + Inverse

(b) Power vs π: OPT + Inverse

*Figure 11.* **Sparsity Impact for OPT-1.3B (Inverse Transform).**

(a) FDR vs π: Qwen + Inverse

(b) Power vs π: Qwen + Inverse

*Figure 12.* **Sparsity Impact for Qwen-2.5-3B (Inverse Transform).**

(a) FDR vs π: Llama + Inverse

(b) Power vs π: Llama + Inverse

*Figure 13.* **Sparsity Impact for Sheared-LLaMA-2.7B (Inverse Transform).**

## B.4. Temperature Impact

We evaluate the robustness of LORD-GoF across varying generation temperatures $\tau \in \{0.1, \ldots, 0.9\}$ on three models and two watermark schemes, with detailed performance metrics provided in Tables 3–7. Across all configurations, Naive methods consistently fail to control FDR—often exceeding $0.40$—rendering them unusable. In contrast, LORD-GoF substantially suppresses FDR relative to the Naive baseline and, in each setting, yields FDR-valid statistics with strong power. The best statistic is setting-dependent: low-temperature regimes often favor spacing/binning-style statistics (such as **Neyman**), while medium- and high-temperature regimes may favor tail- or distribution-sensitive tests such as **Anderson–Darling**, **Watson**, **Kuiper**, or **Chi-squared**, depending on the model and watermark scheme.

*Table 3.* Detection performance on **OPT-1.3B** with **Gumbel-Max** watermark.

| Method | $\tau = 0.1$ | | $\tau = 0.3$ | | $\tau = 0.5$ | | $\tau = 0.7$ | | $\tau = 0.9$ | |
|---|---|---|---|---|---|---|---|---|---|---|
| | FDR | Power | FDR | Power | FDR | Power | FDR | Power | FDR | Power |
| *Offline (Naive-Fixed, $\alpha = 0.05$)* | | | | | | | | | | |
| Naive-Kol | 0.663 | 0.647 | 0.558 | 0.974 | 0.469 | 0.981 | 0.440 | 1.000 | 0.537 | 1.000 |
| Naive-Kui | 0.615 | 0.725 | 0.550 | 0.923 | 0.405 | 0.962 | 0.440 | 1.000 | 0.545 | 1.000 |
| Naive-And | 0.650 | 0.549 | 0.596 | 0.974 | 0.416 | 1.000 | 0.472 | 1.000 | 0.541 | 1.000 |
| Naive-Cra | 0.687 | 0.510 | 0.613 | 0.923 | 0.435 | 1.000 | 0.456 | 1.000 | 0.550 | 1.000 |
| Naive-Wat | 0.660 | 0.647 | 0.598 | 0.846 | 0.451 | 0.962 | 0.423 | 1.000 | 0.550 | 1.000 |
| Naive-Chi | 0.620 | 0.686 | 0.478 | 0.923 | 0.469 | 0.981 | 0.451 | 1.000 | 0.500 | 1.000 |
| Naive-Ney | 0.519 | 0.980 | 0.536 | 1.000 | 0.564 | 0.923 | 0.491 | 1.000 | 0.510 | 1.000 |
| Naive-Phi | 0.525 | 0.941 | 0.514 | 0.923 | 0.521 | 0.865 | 0.466 | 0.982 | 0.546 | 0.980 |
| *Online (LORD-GoF, $\alpha = 0.05$)* | | | | | | | | | | |
| LORD-Kol | 0.000 | 0.137 | 0.000 | 0.436 | 0.021 | 0.885 | **0.000** | **1.000** | 0.091 | 1.000 |
| LORD-Kui | 0.000 | 0.235 | 0.000 | 0.410 | 0.000 | 0.846 | 0.067 | 1.000 | 0.057 | 1.000 |
| LORD-And | 0.000 | 0.137 | 0.000 | 0.538 | **0.038** | **0.981** | 0.018 | 1.000 | 0.091 | 1.000 |
| LORD-Cra | 0.000 | 0.137 | 0.000 | 0.513 | 0.059 | 0.923 | 0.018 | 1.000 | 0.091 | 1.000 |
| LORD-Wat | 0.091 | 0.196 | 0.000 | 0.385 | 0.048 | 0.769 | 0.034 | 1.000 | 0.058 | 0.980 |
| LORD-Chi | 0.095 | 0.373 | 0.000 | 0.513 | 0.000 | 0.885 | 0.034 | 1.000 | 0.091 | 1.000 |
| LORD-Ney | **0.000** | **0.941** | **0.000** | **0.821** | 0.029 | 0.635 | 0.107 | 0.893 | **0.000** | **0.960** |
| LORD-Phi | 0.000 | 0.882 | 0.033 | 0.744 | 0.000 | 0.212 | 0.029 | 0.589 | 0.022 | 0.900 |

## B.5. Robustness against Attack Types

We evaluate the robustness of LORD-GoF against three types of watermark attacks: **Substitution**, **Deletion**, and **Insertion**, with corruption ratios of $r \in \{0.1, 0.2\}$ across three models. The detailed results are summarized in Tables 8–13. Consistent with previous findings, Naive-Fixed methods fail catastrophically, with FDR frequently exceeding $0.50$ across all scenarios.

*Table 4.* Detection performance on **OPT-1.3B** with **Inverse Transform** watermark.

| Method | $\tau = 0.1$ | | $\tau = 0.3$ | | $\tau = 0.5$ | | $\tau = 0.7$ | | $\tau = 0.9$ | |
|---|---|---|---|---|---|---|---|---|---|---|
| | FDR | Power | FDR | Power | FDR | Power | FDR | Power | FDR | Power |
| *Offline (Naive-Fixed, $\alpha = 0.05$)* | | | | | | | | | | |
| Naive-Kol | 0.581 | 0.765 | 0.593 | 0.949 | 0.452 | 0.981 | 0.434 | 1.000 | 0.479 | 1.000 |
| Naive-Kui | 0.521 | 0.882 | 0.575 | 0.949 | 0.441 | 1.000 | 0.434 | 1.000 | 0.462 | 1.000 |
| Naive-And | 0.590 | 0.667 | 0.551 | 0.897 | 0.416 | 1.000 | 0.378 | 1.000 | 0.462 | 1.000 |
| Naive-Cra | 0.581 | 0.608 | 0.582 | 0.846 | 0.400 | 0.981 | 0.404 | 1.000 | 0.468 | 1.000 |
| Naive-Wat | 0.562 | 0.765 | 0.543 | 0.949 | 0.457 | 0.981 | 0.434 | 1.000 | 0.515 | 1.000 |
| Naive-Chi | 0.500 | 0.882 | 0.554 | 0.949 | 0.469 | 1.000 | 0.440 | 1.000 | 0.405 | 1.000 |
| Naive-Ney | 0.414 | 1.000 | 0.552 | 1.000 | 0.480 | 0.981 | 0.495 | 0.875 | 0.485 | 1.000 |
| Naive-Phi | 0.414 | 1.000 | 0.500 | 1.000 | 0.433 | 0.981 | 0.475 | 0.929 | 0.490 | 1.000 |
| *Online (LORD-GoF, $\alpha = 0.05$)* | | | | | | | | | | |
| LORD-Kol | 0.000 | 0.412 | 0.000 | 0.436 | 0.000 | 0.808 | **0.000** | **1.000** | 0.038 | 1.000 |
| LORD-Kui | 0.029 | 0.667 | 0.000 | 0.718 | 0.000 | 0.673 | 0.035 | 0.982 | **0.000** | **1.000** |
| LORD-And | 0.000 | 0.333 | 0.000 | 0.513 | 0.023 | 0.808 | **0.000** | **1.000** | 0.020 | 1.000 |
| LORD-Cra | 0.000 | 0.294 | 0.000 | 0.436 | 0.000 | 0.788 | **0.000** | **1.000** | 0.020 | 1.000 |
| LORD-Wat | 0.000 | 0.529 | 0.000 | 0.462 | 0.000 | 0.538 | 0.070 | 0.946 | 0.020 | 1.000 |
| LORD-Chi | 0.049 | 0.765 | 0.000 | 0.846 | 0.024 | 0.769 | 0.000 | 0.946 | **0.000** | **1.000** |
| LORD-Ney | **0.000** | **1.000** | **0.026** | **0.974** | **0.000** | **0.904** | 0.000 | 0.696 | 0.024 | 0.800 |
| LORD-Phi | 0.019 | 1.000 | 0.051 | 0.949 | 0.061 | 0.885 | 0.028 | 0.625 | 0.000 | 0.860 |

*Table 5.* Detection performance on **Qwen-2.5-3B** with **Gumbel-Max** watermark.

| Method | $\tau = 0.1$ | | $\tau = 0.3$ | | $\tau = 0.5$ | | $\tau = 0.7$ | | $\tau = 0.9$ | |
|---|---|---|---|---|---|---|---|---|---|---|
| | FDR | Power | FDR | Power | FDR | Power | FDR | Power | FDR | Power |
| *Offline (Naive-Fixed, $\alpha = 0.05$)* | | | | | | | | | | |
| Naive-Kol | 0.575 | 0.941 | 0.578 | 0.897 | 0.484 | 0.923 | 0.444 | 0.982 | 0.537 | 1.000 |
| Naive-Kui | 0.557 | 0.922 | 0.550 | 0.923 | 0.415 | 0.923 | 0.444 | 0.982 | 0.545 | 1.000 |
| Naive-And | 0.536 | 0.882 | 0.615 | 0.897 | 0.435 | 0.923 | 0.476 | 0.982 | 0.541 | 1.000 |
| Naive-Cra | 0.559 | 0.882 | 0.620 | 0.897 | 0.455 | 0.923 | 0.470 | 0.946 | 0.550 | 1.000 |
| Naive-Wat | 0.582 | 0.902 | 0.576 | 0.923 | 0.466 | 0.904 | 0.432 | 0.964 | 0.550 | 1.000 |
| Naive-Chi | 0.548 | 0.922 | 0.478 | 0.923 | 0.479 | 0.942 | 0.460 | 0.964 | 0.500 | 1.000 |
| Naive-Ney | 0.519 | 0.980 | 0.556 | 0.923 | 0.559 | 0.942 | 0.491 | 1.000 | 0.510 | 1.000 |
| Naive-Phi | 0.525 | 0.941 | 0.514 | 0.923 | 0.516 | 0.885 | 0.462 | 1.000 | 0.541 | 1.000 |
| *Online (LORD-GoF, $\alpha = 0.05$)* | | | | | | | | | | |
| LORD-Kol | 0.043 | 0.863 | 0.000 | 0.846 | 0.000 | 0.769 | **0.000** | **0.929** | 0.093 | 0.980 |
| LORD-Kui | 0.022 | 0.882 | 0.029 | 0.846 | 0.024 | 0.769 | 0.071 | 0.929 | 0.057 | 1.000 |
| LORD-And | 0.083 | 0.863 | 0.000 | 0.846 | 0.047 | 0.788 | 0.019 | 0.911 | 0.091 | 1.000 |
| LORD-Cra | 0.083 | 0.863 | 0.000 | 0.846 | 0.025 | 0.750 | 0.020 | 0.893 | 0.091 | 1.000 |
| LORD-Wat | 0.122 | 0.843 | 0.000 | 0.795 | 0.026 | 0.712 | 0.038 | 0.893 | 0.058 | 0.980 |
| LORD-Chi | 0.043 | 0.882 | 0.029 | 0.872 | 0.000 | 0.846 | 0.037 | 0.929 | 0.091 | 1.000 |
| LORD-Ney | **0.020** | **0.961** | **0.000** | **0.897** | **0.022** | **0.865** | 0.115 | 0.964 | **0.000** | **1.000** |
| LORD-Phi | 0.000 | 0.941 | 0.029 | 0.872 | 0.048 | 0.769 | 0.073 | 0.911 | 0.020 | 0.960 |

*Table 6.* Detection performance on **Sheared-LLaMA-2.7B** with **Inverse Transform** watermark.

| Method | $\tau = 0.1$ | | $\tau = 0.3$ | | $\tau = 0.5$ | | $\tau = 0.7$ | | $\tau = 0.9$ | |
|---|---|---|---|---|---|---|---|---|---|---|
| | FDR | Power | FDR | Power | FDR | Power | FDR | Power | FDR | Power |
| *Offline (Naive-Fixed, $\alpha = 0.05$)* | | | | | | | | | | |
| Naive-Kol | 0.593 | 0.725 | 0.614 | 0.872 | 0.467 | 0.923 | 0.434 | 1.000 | 0.479 | 1.000 |
| Naive-Kui | 0.544 | 0.804 | 0.581 | 0.923 | 0.466 | 0.904 | 0.434 | 1.000 | 0.462 | 1.000 |
| Naive-And | 0.598 | 0.647 | 0.573 | 0.821 | 0.435 | 0.923 | 0.378 | 1.000 | 0.462 | 1.000 |
| Naive-Cra | 0.581 | 0.608 | 0.597 | 0.795 | 0.420 | 0.904 | 0.404 | 1.000 | 0.468 | 1.000 |
| Naive-Wat | 0.575 | 0.725 | 0.579 | 0.821 | 0.506 | 0.808 | 0.434 | 1.000 | 0.515 | 1.000 |
| Naive-Chi | 0.529 | 0.784 | 0.548 | 0.974 | 0.495 | 0.904 | 0.444 | 0.982 | 0.405 | 1.000 |
| Naive-Ney | 0.414 | 1.000 | 0.552 | 1.000 | 0.490 | 0.942 | 0.522 | 0.786 | 0.495 | 0.960 |
| Naive-Phi | 0.419 | 0.980 | 0.506 | 0.974 | 0.448 | 0.923 | 0.500 | 0.839 | 0.500 | 0.960 |
| *Online (LORD-GoF, $\alpha = 0.05$)* | | | | | | | | | | |
| LORD-Kol | 0.111 | 0.157 | 0.000 | 0.564 | 0.000 | 0.558 | **0.000** | **0.964** | 0.038 | 1.000 |
| LORD-Kui | 0.074 | 0.490 | 0.000 | 0.667 | 0.034 | 0.538 | 0.037 | 0.929 | **0.000** | **1.000** |
| LORD-And | 0.000 | 0.176 | 0.000 | 0.462 | 0.031 | 0.596 | **0.000** | **0.964** | 0.020 | 1.000 |
| LORD-Cra | 0.000 | 0.078 | 0.000 | 0.410 | 0.000 | 0.577 | **0.000** | **0.964** | 0.020 | 1.000 |
| LORD-Wat | 0.059 | 0.314 | 0.000 | 0.436 | 0.043 | 0.423 | 0.075 | 0.875 | 0.020 | 1.000 |
| LORD-Chi | 0.000 | 0.627 | 0.000 | 0.692 | 0.030 | 0.615 | 0.000 | 0.893 | **0.000** | **1.000** |
| LORD-Ney | **0.000** | **1.000** | 0.026 | 0.974 | **0.000** | **0.827** | 0.040 | 0.429 | 0.000 | 0.360 |
| LORD-Phi | 0.020 | 0.941 | 0.053 | 0.923 | 0.048 | 0.769 | 0.087 | 0.375 | 0.000 | 0.540 |

*Table 7.* Detection performance on **Qwen-2.5-3B** with **Inverse Transform** watermark.

| Method | $\tau = 0.1$ | | $\tau = 0.3$ | | $\tau = 0.5$ | | $\tau = 0.7$ | | $\tau = 0.9$ | |
|---|---|---|---|---|---|---|---|---|---|---|
| | FDR | Power | FDR | Power | FDR | Power | FDR | Power | FDR | Power |
| *Offline (Naive-Fixed, $\alpha = 0.05$)* | | | | | | | | | | |
| Naive-Kol | 0.607 | 0.686 | 0.643 | 0.769 | 0.462 | 0.942 | 0.443 | 0.964 | 0.479 | 1.000 |
| Naive-Kui | 0.538 | 0.824 | 0.625 | 0.769 | 0.461 | 0.923 | 0.439 | 0.982 | 0.462 | 1.000 |
| Naive-And | 0.598 | 0.647 | 0.589 | 0.769 | 0.425 | 0.962 | 0.378 | 1.000 | 0.462 | 1.000 |
| Naive-Cra | 0.589 | 0.588 | 0.630 | 0.692 | 0.415 | 0.923 | 0.413 | 0.964 | 0.468 | 1.000 |
| Naive-Wat | 0.575 | 0.725 | 0.595 | 0.769 | 0.494 | 0.846 | 0.439 | 0.982 | 0.515 | 1.000 |
| Naive-Chi | 0.506 | 0.863 | 0.597 | 0.795 | 0.495 | 0.904 | 0.444 | 0.982 | 0.405 | 1.000 |
| Naive-Ney | 0.414 | 1.000 | 0.578 | 0.897 | 0.490 | 0.942 | 0.490 | 0.893 | 0.490 | 0.980 |
| Naive-Phi | 0.414 | 1.000 | 0.527 | 0.897 | 0.459 | 0.885 | 0.490 | 0.875 | 0.495 | 0.980 |
| *Online (LORD-GoF, $\alpha = 0.05$)* | | | | | | | | | | |
| LORD-Kol | 0.000 | 0.314 | 0.000 | 0.333 | 0.000 | 0.577 | 0.000 | 0.911 | 0.038 | 1.000 |
| LORD-Kui | 0.000 | 0.549 | 0.000 | 0.538 | 0.000 | 0.538 | 0.021 | 0.839 | **0.000** | **1.000** |
| LORD-And | 0.000 | 0.294 | 0.091 | 0.256 | 0.029 | 0.635 | **0.000** | **0.929** | 0.020 | 1.000 |
| LORD-Cra | 0.000 | 0.255 | 0.000 | 0.231 | 0.000 | 0.654 | **0.000** | **0.929** | 0.020 | 1.000 |
| LORD-Wat | 0.000 | 0.412 | 0.000 | 0.462 | 0.000 | 0.442 | 0.064 | 0.786 | 0.020 | 1.000 |
| LORD-Chi | 0.032 | 0.588 | 0.000 | 0.615 | 0.030 | 0.615 | 0.000 | 0.804 | **0.000** | **1.000** |
| LORD-Ney | **0.000** | **1.000** | 0.028 | 0.897 | **0.000** | **0.865** | 0.029 | 0.589 | 0.023 | 0.860 |
| LORD-Phi | 0.019 | 1.000 | 0.056 | 0.872 | 0.024 | 0.788 | 0.028 | 0.625 | 0.000 | 0.880 |

*Table 8.* Robustness of **Gumbel-Max** (OPT-1.3B) against Substitution, Deletion, and Insertion. Values are reported as FDR / Power. **Bold** indicates the highest Power among methods with FDR $\leq 0.05$.

| Method | Subst ($r = 0.1$) | | Subst ($r = 0.2$) | | Del ($r = 0.1$) | | Del ($r = 0.2$) | | Ins ($r = 0.1$) | | Ins ($r = 0.2$) | |
|---|---|---|---|---|---|---|---|---|---|---|---|---|
| | FDR | Power | FDR | Power | FDR | Power | FDR | Power | FDR | Power | FDR | Power |
| *Offline (Naive-Fixed, $\alpha = 0.05$)* | | | | | | | | | | | | |
| Naive-Kol | 0.525 | 0.979 | 0.589 | 1.000 | 0.500 | 1.000 | 0.509 | 1.000 | 0.545 | 1.000 | 0.552 | 1.000 |
| Naive-Kui | 0.525 | 0.979 | 0.526 | 1.000 | 0.473 | 1.000 | 0.439 | 0.982 | 0.527 | 0.978 | 0.517 | 1.000 |
| Naive-Cra | 0.544 | 0.979 | 0.549 | 1.000 | 0.500 | 1.000 | 0.467 | 1.000 | 0.511 | 1.000 | 0.522 | 1.000 |
| Naive-And | 0.530 | 0.979 | 0.545 | 1.000 | 0.489 | 1.000 | 0.456 | 1.000 | 0.500 | 1.000 | 0.522 | 1.000 |
| Naive-Wat | 0.565 | 0.979 | 0.500 | 0.978 | 0.478 | 1.000 | 0.450 | 0.982 | 0.494 | 0.978 | 0.538 | 1.000 |
| Naive-Chi | 0.520 | 0.979 | 0.511 | 1.000 | 0.484 | 1.000 | 0.446 | 1.000 | 0.511 | 1.000 | 0.543 | 1.000 |
| Naive-Ney | 0.520 | 1.000 | 0.505 | 1.000 | 0.598 | 0.979 | 0.604 | 0.982 | 0.559 | 1.000 | 0.547 | 1.000 |
| Naive-Phi | 0.534 | 1.000 | 0.476 | 0.957 | 0.553 | 0.958 | 0.522 | 0.982 | 0.541 | 1.000 | 0.476 | 1.000 |
| *Online (LORD-GoF, Target $\alpha = 0.05$)* | | | | | | | | | | | | |
| LORD-Kol | 0.096 | 0.979 | **0.000** | **0.957** | **0.000** | **1.000** | 0.035 | 0.982 | **0.022** | **0.978** | **0.000** | **1.000** |
| LORD-Kui | 0.096 | 0.979 | 0.023 | 0.935 | 0.059 | 1.000 | 0.018 | 0.982 | 0.064 | 0.978 | 0.000 | 0.977 |
| LORD-Cra | 0.078 | 0.979 | 0.000 | 0.935 | 0.020 | 1.000 | **0.018** | **1.000** | 0.065 | 0.956 | 0.070 | 0.930 |
| LORD-And | 0.078 | 0.979 | 0.000 | 0.957 | **0.000** | **1.000** | **0.018** | **1.000** | 0.064 | 0.978 | 0.106 | 0.977 |
| LORD-Wat | 0.062 | 0.938 | 0.024 | 0.891 | 0.061 | 0.958 | 0.019 | 0.911 | 0.064 | 0.978 | 0.000 | 0.930 |
| LORD-Chi | 0.096 | 0.979 | 0.022 | 0.957 | 0.077 | 1.000 | 0.018 | 0.982 | 0.000 | 0.956 | 0.044 | 1.000 |
| LORD-Ney | **0.022** | **0.938** | 0.026 | 0.826 | 0.045 | 0.875 | 0.103 | 0.929 | 0.024 | 0.889 | 0.024 | 0.930 |
| LORD-Phi | 0.071 | 0.812 | 0.057 | 0.717 | 0.028 | 0.729 | 0.038 | 0.893 | 0.029 | 0.733 | 0.027 | 0.837 |

*Table 9.* Robustness of **Inverse Transform** (OPT-1.3B) against Substitution, Deletion, and Insertion. Values are reported as FDR / Power. **Bold** indicates the highest Power among methods with FDR $\leq 0.05$.

| Method | Subst ($r = 0.1$) | | Subst ($r = 0.2$) | | Del ($r = 0.1$) | | Del ($r = 0.2$) | | Ins ($r = 0.1$) | | Ins ($r = 0.2$) | |
|---|---|---|---|---|---|---|---|---|---|---|---|---|
| | FDR | Power | FDR | Power | FDR | Power | FDR | Power | FDR | Power | FDR | Power |
| *Offline (Naive-Fixed, $\alpha = 0.05$)* | | | | | | | | | | | | |
| Naive-Kol | 0.484 | 1.000 | 0.545 | 1.000 | 0.484 | 1.000 | 0.446 | 1.000 | 0.591 | 1.000 | 0.442 | 1.000 |
| Naive-Kui | 0.505 | 1.000 | 0.549 | 1.000 | 0.534 | 1.000 | 0.477 | 1.000 | 0.554 | 1.000 | 0.522 | 1.000 |
| Naive-Cra | 0.505 | 0.958 | 0.511 | 0.978 | 0.505 | 1.000 | 0.385 | 1.000 | 0.595 | 1.000 | 0.462 | 0.977 |
| Naive-And | 0.489 | 0.979 | 0.494 | 0.978 | 0.525 | 1.000 | 0.404 | 1.000 | 0.587 | 1.000 | 0.481 | 0.977 |
| Naive-Wat | 0.505 | 0.979 | 0.531 | 0.978 | 0.561 | 0.979 | 0.446 | 1.000 | 0.567 | 1.000 | 0.500 | 1.000 |
| Naive-Chi | 0.460 | 0.979 | 0.465 | 1.000 | 0.505 | 1.000 | 0.461 | 0.982 | 0.602 | 1.000 | 0.533 | 1.000 |
| Naive-Ney | 0.435 | 1.000 | 0.551 | 0.957 | 0.610 | 1.000 | 0.631 | 0.982 | 0.554 | 1.000 | 0.598 | 1.000 |
| Naive-Phi | 0.505 | 1.000 | 0.494 | 0.957 | 0.505 | 1.000 | 0.517 | 1.000 | 0.541 | 1.000 | 0.566 | 1.000 |
| *Online (LORD-GoF, Target $\alpha = 0.05$)* | | | | | | | | | | | | |
| LORD-Kol | 0.045 | 0.875 | 0.023 | 0.935 | 0.042 | 0.958 | 0.086 | 0.946 | 0.118 | 1.000 | 0.000 | 0.907 |
| LORD-Kui | 0.043 | 0.917 | 0.000 | 0.870 | 0.042 | 0.958 | 0.119 | 0.929 | 0.062 | 1.000 | 0.000 | 0.907 |
| LORD-Cra | 0.065 | 0.896 | 0.024 | 0.870 | 0.022 | 0.938 | 0.073 | 0.911 | 0.104 | 0.956 | 0.000 | 0.884 |
| LORD-And | 0.043 | 0.917 | 0.000 | 0.891 | 0.022 | 0.938 | 0.103 | 0.929 | 0.102 | 0.978 | 0.000 | 0.930 |
| LORD-Wat | 0.025 | 0.812 | 0.000 | 0.761 | 0.000 | 0.958 | 0.107 | 0.893 | 0.065 | 0.956 | 0.000 | 0.860 |
| LORD-Chi | 0.044 | 0.896 | 0.045 | 0.913 | **0.000** | **0.979** | **0.000** | **0.946** | **0.000** | **1.000** | 0.024 | 0.953 |
| LORD-Ney | **0.021** | **0.958** | 0.064 | 0.957 | 0.130 | 0.979 | 0.182 | 0.964 | 0.022 | 1.000 | **0.044** | **1.000** |
| LORD-Phi | 0.043 | 0.938 | **0.000** | **0.935** | 0.000 | 0.938 | 0.000 | 0.893 | 0.022 | 1.000 | 0.065 | 1.000 |

*Table 10.* Robustness of **Gumbel-Max** (Qwen-2.5-3B) against Substitution, Deletion, and Insertion. Values are reported as FDR / Power. **Bold** indicates the highest Power among methods with $\text{FDR} \leq 0.05$.

| Method | Subst ($r = 0.1$) | | Subst ($r = 0.2$) | | Del ($r = 0.1$) | | Del ($r = 0.2$) | | Ins ($r = 0.1$) | | Ins ($r = 0.2$) | |
|---|---|---|---|---|---|---|---|---|---|---|---|---|
| | FDR | Power | FDR | Power | FDR | Power | FDR | Power | FDR | Power | FDR | Power |
| *Offline (Naive-Fixed, $\alpha = 0.05$)* | | | | | | | | | | | | |
| Naive-Kol | 0.542 | 0.917 | 0.595 | 0.978 | 0.516 | 0.938 | 0.518 | 0.964 | 0.545 | 1.000 | 0.564 | 0.953 |
| Naive-Kui | 0.542 | 0.917 | 0.537 | 0.957 | 0.494 | 0.917 | 0.443 | 0.964 | 0.521 | 1.000 | 0.523 | 0.977 |
| Naive-Cra | 0.571 | 0.875 | 0.566 | 0.935 | 0.516 | 0.938 | 0.480 | 0.946 | 0.511 | 1.000 | 0.528 | 0.977 |
| Naive-And | 0.546 | 0.917 | 0.556 | 0.957 | 0.500 | 0.958 | 0.465 | 0.964 | 0.500 | 1.000 | 0.528 | 0.977 |
| Naive-Wat | 0.587 | 0.896 | 0.517 | 0.913 | 0.500 | 0.917 | 0.455 | 0.964 | 0.494 | 0.978 | 0.543 | 0.977 |
| Naive-Chi | 0.537 | 0.917 | 0.522 | 0.957 | 0.495 | 0.958 | 0.455 | 0.964 | 0.511 | 1.000 | 0.554 | 0.953 |
| Naive-Ney | 0.536 | 0.938 | 0.522 | 0.935 | 0.603 | 0.958 | 0.604 | 0.982 | 0.559 | 1.000 | 0.553 | 0.977 |
| Naive-Phi | 0.550 | 0.938 | 0.482 | 0.935 | 0.548 | 0.979 | 0.526 | 0.964 | 0.541 | 1.000 | 0.481 | 0.977 |
| *Online (LORD-GOF, Target $\alpha = 0.05$)* | | | | | | | | | | | | |
| LORD-Kol | 0.095 | 0.792 | 0.000 | 0.848 | 0.000 | 0.875 | 0.039 | 0.875 | 0.023 | 0.933 | 0.000 | 0.953 |
| LORD-Kui | 0.111 | 0.833 | 0.000 | 0.870 | 0.045 | 0.875 | 0.020 | 0.875 | 0.067 | 0.933 | 0.000 | 0.953 |
| LORD-Cra | 0.071 | 0.812 | 0.000 | 0.848 | 0.025 | 0.812 | 0.020 | 0.893 | **0.043** | **0.978** | 0.068 | 0.953 |
| LORD-And | 0.068 | 0.854 | 0.024 | 0.891 | 0.000 | 0.875 | 0.020 | 0.893 | 0.062 | 1.000 | 0.109 | 0.953 |
| LORD-Wat | **0.050** | **0.792** | 0.026 | 0.826 | 0.068 | 0.854 | 0.021 | 0.839 | 0.067 | 0.933 | 0.000 | 0.930 |
| LORD-Chi | 0.089 | 0.854 | **0.000** | **0.891** | 0.083 | 0.917 | **0.019** | **0.911** | 0.000 | 0.956 | 0.047 | 0.953 |
| LORD-Ney | 0.062 | 0.938 | 0.024 | 0.891 | 0.061 | 0.958 | 0.071 | 0.929 | **0.043** | **0.978** | **0.045** | **0.977** |
| LORD-Phi | 0.062 | 0.938 | 0.049 | 0.848 | **0.023** | **0.896** | 0.020 | 0.875 | 0.044 | 0.956 | 0.024 | 0.930 |

*Table 11.* Robustness of **Inverse Transform** (Qwen-2.5-3B) against Substitution, Deletion, and Insertion. Values are reported as FDR / Power. **Bold** indicates the highest Power among methods with $\text{FDR} \leq 0.05$.

| Method | Subst ($r = 0.1$) | | Subst ($r = 0.2$) | | Del ($r = 0.1$) | | Del ($r = 0.2$) | | Ins ($r = 0.1$) | | Ins ($r = 0.2$) | |
|---|---|---|---|---|---|---|---|---|---|---|---|---|
| | FDR | Power | FDR | Power | FDR | Power | FDR | Power | FDR | Power | FDR | Power |
| *Offline (Naive-Fixed, $\alpha = 0.05$)* | | | | | | | | | | | | |
| Naive-Kol | 0.500 | 0.938 | 0.550 | 0.978 | 0.495 | 0.958 | 0.450 | 0.982 | 0.596 | 0.978 | 0.447 | 0.977 |
| Naive-Kui | 0.516 | 0.958 | 0.549 | 1.000 | 0.545 | 0.958 | 0.486 | 0.964 | 0.560 | 0.978 | 0.540 | 0.930 |
| Naive-Cra | 0.511 | 0.938 | 0.511 | 0.978 | 0.516 | 0.958 | 0.389 | 0.982 | 0.600 | 0.978 | 0.468 | 0.953 |
| Naive-And | 0.500 | 0.938 | 0.494 | 0.978 | 0.530 | 0.979 | 0.404 | 1.000 | 0.593 | 0.978 | 0.487 | 0.953 |
| Naive-Wat | 0.516 | 0.938 | 0.526 | 1.000 | 0.566 | 0.958 | 0.464 | 0.929 | 0.578 | 0.956 | 0.518 | 0.930 |
| Naive-Chi | 0.471 | 0.938 | 0.465 | 1.000 | 0.516 | 0.958 | 0.456 | 1.000 | 0.613 | 0.956 | 0.551 | 0.930 |
| Naive-Ney | 0.457 | 0.917 | 0.551 | 0.957 | 0.615 | 0.979 | 0.635 | 0.964 | 0.560 | 0.978 | 0.615 | 0.930 |
| Naive-Phi | 0.527 | 0.917 | 0.489 | 0.978 | 0.516 | 0.958 | 0.526 | 0.964 | 0.552 | 0.956 | 0.583 | 0.930 |
| *Online (LORD-GOF, Target $\alpha = 0.05$)* | | | | | | | | | | | | |
| LORD-Kol | 0.053 | 0.750 | 0.000 | 0.826 | 0.043 | 0.917 | 0.078 | 0.839 | 0.128 | 0.911 | 0.000 | 0.767 |
| LORD-Kui | 0.027 | 0.750 | 0.000 | 0.804 | 0.045 | 0.875 | 0.120 | 0.786 | 0.070 | 0.889 | 0.000 | 0.698 |
| LORD-Cra | 0.027 | 0.750 | 0.024 | 0.870 | **0.022** | **0.917** | 0.082 | 0.804 | 0.116 | 0.844 | 0.000 | 0.814 |
| LORD-And | 0.025 | 0.812 | **0.000** | **0.870** | **0.022** | **0.917** | 0.115 | 0.821 | 0.091 | 0.889 | 0.000 | 0.837 |
| LORD-Wat | 0.032 | 0.625 | 0.000 | 0.739 | 0.000 | 0.812 | 0.103 | 0.625 | 0.077 | 0.800 | 0.000 | 0.581 |
| LORD-Chi | 0.048 | 0.833 | 0.024 | 0.870 | 0.000 | 0.875 | **0.000** | **0.821** | 0.000 | 0.889 | 0.000 | 0.791 |
| LORD-Ney | **0.000** | **0.875** | 0.065 | 0.935 | 0.143 | 0.875 | 0.197 | 0.875 | **0.023** | **0.933** | **0.049** | **0.907** |
| LORD-Phi | 0.047 | 0.854 | **0.000** | **0.870** | 0.000 | 0.833 | 0.000 | 0.804 | 0.000 | 0.889 | 0.077 | 0.837 |

*Table 12.* Robustness of **Gumbel-Max** (Sheared-LLaMA-2.7B) against Substitution, Deletion, and Insertion. Values are reported as FDR / Power. **Bold** indicates the highest Power among methods with $\text{FDR} \leq 0.05$.

| Method | Subst ($r = 0.1$) | | Subst ($r = 0.2$) | | Del ($r = 0.1$) | | Del ($r = 0.2$) | | Ins ($r = 0.1$) | | Ins ($r = 0.2$) | |
|---|---|---|---|---|---|---|---|---|---|---|---|---|
| | FDR | Power | FDR | Power | FDR | Power | FDR | Power | FDR | Power | FDR | Power |
| *Offline (Naive-Fixed, $\alpha = 0.05$)* | | | | | | | | | | | | |
| Naive-Kol | 0.536 | 0.938 | 0.611 | 0.913 | 0.511 | 0.958 | 0.527 | 0.929 | 0.562 | 0.933 | 0.558 | 0.977 |
| Naive-Kui | 0.531 | 0.958 | 0.548 | 0.913 | 0.483 | 0.958 | 0.453 | 0.929 | 0.538 | 0.933 | 0.523 | 0.977 |
| Naive-Cra | 0.560 | 0.917 | 0.577 | 0.891 | 0.511 | 0.958 | 0.485 | 0.929 | 0.534 | 0.911 | 0.528 | 0.977 |
| Naive-And | 0.546 | 0.917 | 0.567 | 0.913 | 0.500 | 0.958 | 0.475 | 0.929 | 0.517 | 0.933 | 0.528 | 0.977 |
| Naive-Wat | 0.581 | 0.917 | 0.523 | 0.891 | 0.494 | 0.938 | 0.469 | 0.911 | 0.506 | 0.933 | 0.543 | 0.977 |
| Naive-Chi | 0.520 | 0.979 | 0.527 | 0.935 | 0.495 | 0.958 | 0.464 | 0.929 | 0.528 | 0.933 | 0.548 | 0.977 |
| Naive-Ney | 0.536 | 0.938 | 0.522 | 0.935 | 0.603 | 0.958 | 0.622 | 0.911 | 0.582 | 0.911 | 0.553 | 0.977 |
| Naive-Phi | 0.550 | 0.938 | 0.482 | 0.935 | 0.553 | 0.958 | 0.541 | 0.911 | 0.564 | 0.911 | 0.481 | 0.977 |
| *Online (LORD-GoF, Target $\alpha = 0.05$)* | | | | | | | | | | | | |
| LORD-Kol | 0.102 | 0.917 | 0.000 | 0.804 | 0.000 | 0.917 | 0.039 | 0.875 | 0.024 | 0.911 | 0.000 | 0.930 |
| LORD-Kui | 0.083 | 0.917 | 0.027 | 0.783 | 0.022 | 0.917 | 0.021 | 0.839 | 0.068 | 0.911 | 0.000 | 0.907 |
| LORD-Cra | 0.083 | 0.917 | 0.000 | 0.826 | 0.022 | 0.938 | 0.021 | 0.839 | 0.068 | 0.911 | 0.070 | 0.930 |
| LORD-And | 0.083 | 0.917 | 0.026 | 0.826 | **0.000** | **0.938** | 0.020 | 0.857 | 0.068 | 0.911 | 0.106 | 0.977 |
| LORD-Wat | 0.067 | 0.875 | 0.000 | 0.674 | 0.024 | 0.854 | 0.000 | 0.804 | 0.068 | 0.911 | 0.000 | 0.907 |
| LORD-Chi | 0.064 | 0.917 | 0.025 | 0.848 | 0.082 | 0.938 | **0.020** | **0.875** | **0.000** | **0.911** | **0.045** | **0.977** |
| LORD-Ney | **0.022** | **0.938** | **0.024** | **0.891** | 0.064 | 0.917 | 0.075 | 0.875 | 0.047 | 0.911 | **0.045** | **0.977** |
| LORD-Phi | 0.043 | 0.917 | 0.053 | 0.783 | 0.043 | 0.917 | 0.040 | 0.857 | 0.047 | 0.911 | 0.024 | 0.953 |

*Table 13.* Robustness of **Inverse Transform** (Sheared-LLaMA-2.7B) against Substitution, Deletion, and Insertion. Values are reported as FDR / Power. **Bold** indicates the highest Power among methods with $\text{FDR} \leq 0.05$.

| Method | Subst ($r = 0.1$) | | Subst ($r = 0.2$) | | Del ($r = 0.1$) | | Del ($r = 0.2$) | | Ins ($r = 0.1$) | | Ins ($r = 0.2$) | |
|---|---|---|---|---|---|---|---|---|---|---|---|---|
| | FDR | Power | FDR | Power | FDR | Power | FDR | Power | FDR | Power | FDR | Power |
| *Offline (Naive-Fixed, $\alpha = 0.05$)* | | | | | | | | | | | | |
| Naive-Kol | 0.489 | 0.979 | 0.567 | 0.913 | 0.489 | 0.979 | 0.450 | 0.982 | 0.591 | 1.000 | 0.447 | 0.977 |
| Naive-Kui | 0.510 | 0.979 | 0.589 | 0.848 | 0.539 | 0.979 | 0.481 | 0.982 | 0.554 | 1.000 | 0.528 | 0.977 |
| Naive-Cra | 0.500 | 0.979 | 0.534 | 0.891 | 0.516 | 0.958 | 0.393 | 0.964 | 0.595 | 1.000 | 0.462 | 0.977 |
| Naive-And | 0.489 | 0.979 | 0.506 | 0.935 | 0.530 | 0.979 | 0.409 | 0.982 | 0.587 | 1.000 | 0.481 | 0.977 |
| Naive-Wat | 0.516 | 0.938 | 0.567 | 0.848 | 0.571 | 0.938 | 0.450 | 0.982 | 0.573 | 0.978 | 0.512 | 0.953 |
| Naive-Chi | 0.460 | 0.979 | 0.482 | 0.935 | 0.510 | 0.979 | 0.461 | 0.982 | 0.602 | 1.000 | 0.538 | 0.977 |
| Naive-Ney | 0.440 | 0.979 | 0.545 | 0.978 | 0.620 | 0.958 | 0.631 | 0.982 | 0.560 | 0.978 | 0.610 | 0.953 |
| Naive-Phi | 0.516 | 0.958 | 0.500 | 0.935 | 0.521 | 0.938 | 0.522 | 0.982 | 0.546 | 0.978 | 0.577 | 0.953 |
| *Online (LORD-GoF, Target $\alpha = 0.05$)* | | | | | | | | | | | | |
| LORD-Kol | 0.050 | 0.792 | 0.033 | 0.630 | 0.049 | 0.812 | 0.070 | 0.946 | 0.122 | 0.956 | 0.000 | 0.814 |
| LORD-Kui | 0.031 | 0.646 | 0.000 | 0.674 | 0.024 | 0.833 | 0.119 | 0.929 | 0.045 | 0.933 | 0.000 | 0.791 |
| LORD-Cra | 0.068 | 0.854 | 0.000 | 0.630 | 0.028 | 0.729 | 0.077 | 0.857 | 0.106 | 0.933 | 0.000 | 0.884 |
| LORD-And | 0.049 | 0.812 | 0.000 | 0.652 | 0.025 | 0.812 | 0.111 | 0.857 | 0.106 | 0.933 | 0.000 | 0.884 |
| LORD-Wat | 0.062 | 0.625 | 0.000 | 0.565 | 0.000 | 0.729 | 0.109 | 0.875 | 0.051 | 0.822 | 0.000 | 0.698 |
| LORD-Chi | 0.026 | 0.792 | 0.054 | 0.761 | 0.000 | 0.854 | **0.000** | **0.946** | **0.000** | **0.956** | 0.027 | 0.837 |
| LORD-Ney | **0.023** | **0.875** | **0.045** | **0.913** | 0.137 | 0.917 | 0.172 | 0.946 | 0.023 | 0.956 | **0.024** | **0.930** |
| LORD-Phi | 0.047 | 0.854 | 0.000 | 0.891 | **0.000** | **0.896** | 0.000 | 0.946 | 0.000 | 0.911 | 0.071 | 0.907 |

## C. Hyperparameter Sensitivity Analysis

The performance of the LORD algorithm relies on two key hyperparameters: the initial wealth $W_0$ and the decay exponent $s$ (where the wealth update sequence $\gamma_j \propto j^{-s}$). In this section, we conduct a comprehensive sensitivity analysis across all six model-watermark configurations. To evaluate the robustness of the LORD-GoF *framework* as a whole, we report the **Average FDR** and **Average Power** computed over the full suite of eight GoF statistics.

### C.1. Impact of Initial Wealth ($W_0$)

$W_0$ acts as the "starting budget" for hypothesis testing. We evaluate values ranging from highly conservative (0.0005, i.e., $0.01\alpha$) to aggressive (0.025, i.e., $0.5\alpha$), with our default set to $W_0 = 0.01$ ($0.2\alpha$).

*Table 14.* **Sensitivity to Initial Wealth ($W_0$).** Fixed decay $s = 1.2$. Values represent the average across 8 GoF statistics.

| $W_0$ | OPT+Gum | | OPT+Inv | | Qwen+Gum | | Qwen+Inv | | Llama+Gum | | Llama+Inv | |
|---|---|---|---|---|---|---|---|---|---|---|---|---|
| | FDR | Power | FDR | Power | FDR | Power | FDR | Power | FDR | Power | FDR | Power |
| 0.0005 | 0.068 | 0.816 | 0.046 | 0.916 | 0.063 | 0.853 | 0.047 | 0.823 | 0.065 | 0.795 | 0.037 | 0.770 |
| 0.0010 | 0.066 | 0.817 | 0.046 | 0.921 | 0.061 | 0.851 | 0.047 | 0.823 | 0.065 | 0.795 | 0.043 | 0.774 |
| 0.0050 | 0.052 | 0.819 | 0.041 | 0.925 | 0.055 | 0.850 | 0.043 | 0.823 | 0.058 | 0.793 | 0.037 | 0.772 |
| **0.0100** | **0.048** | **0.818** | **0.036** | **0.923** | **0.050** | **0.848** | **0.036** | **0.816** | **0.045** | **0.791** | **0.029** | **0.774** |
| 0.0250 | 0.031 | 0.803 | 0.024 | 0.910 | 0.036 | 0.842 | 0.023 | 0.800 | 0.032 | 0.782 | 0.022 | 0.774 |

**Analysis.** As shown in Table 14, the choice of $W_0$ involves a trade-off between early discovery speed and long-term stability.

- **Low $W_0$ ($< 0.001$):** While safe, extremely low initial wealth delays the first rejection, slightly reducing overall power (e.g., OPT+Inv Power drops from 0.925 to 0.916).

- **High $W_0$ ($\geq 0.025$):** Surprisingly, very high initial wealth can sometimes *reduce* average power (e.g., Qwen+Inv drops to 0.800). This occurs because a large initial alpha allows early false positives that rapidly deplete wealth, leaving the algorithm "starved" for later true signals.

- **Robust Default:** Our choice of $W_0 = 0.01$ retains near-peak power across all models while yielding tighter FDR control than smaller values (e.g., the average FDR is uniformly lower than at $W_0 = 0.005$ at essentially unchanged power), confirming it as a robust starting point.

### C.2. Impact of Decay Parameter ($s$)

The exponent $s$ controls the "memory" of the wealth distribution. We test $s \in [1.05, 2.0]$, where smaller $s$ implies longer memory (wealth spread further into the future).

*Table 15.* **Sensitivity to Decay Exponent ($s$).** Initial wealth fixed at the default $W_0 = 0.01$; values are averaged across the 8 GoF statistics.

| Gamma ($s$) | OPT+Gum | | OPT+Inv | | Qwen+Gum | | Qwen+Inv | | Llama+Gum | | Llama+Inv | |
|---|---|---|---|---|---|---|---|---|---|---|---|---|
| | FDR | Power | FDR | Power | FDR | Power | FDR | Power | FDR | Power | FDR | Power |
| 1.05 | 0.046 | 0.821 | 0.037 | 0.928 | 0.051 | 0.851 | 0.038 | 0.824 | 0.046 | 0.793 | 0.030 | 0.787 |
| **1.20** | **0.048** | **0.818** | **0.036** | **0.923** | **0.050** | **0.848** | **0.036** | **0.816** | **0.045** | **0.791** | **0.029** | **0.774** |
| 1.40 | 0.049 | 0.809 | 0.037 | 0.916 | 0.054 | 0.847 | 0.039 | 0.808 | 0.049 | 0.787 | 0.034 | 0.765 |
| 1.60 | 0.046 | 0.792 | 0.036 | 0.911 | 0.053 | 0.845 | 0.038 | 0.790 | 0.051 | 0.786 | 0.036 | 0.754 |
| 1.80 | 0.048 | 0.778 | 0.034 | 0.897 | 0.054 | 0.840 | 0.039 | 0.774 | 0.052 | 0.781 | 0.035 | 0.736 |
| 2.00 | 0.053 | 0.762 | 0.032 | 0.894 | 0.057 | 0.835 | 0.037 | 0.755 | 0.053 | 0.767 | 0.033 | 0.718 |

**Analysis.** Table 15 reveals a consistent trend:

- **Long Memory Benefit:** Lower $s$ values $(1.05 - 1.20)$ generally yield higher Power. This is expected in our simulation setting where watermark signals are uniformly distributed (constant sparsity $\pi$). A "long memory" ensures that wealth earned from early detections is preserved for later in the stream.

- **Performance Drop at High $s$:** As $s$ increases to 2.0, the algorithm becomes "short-sighted." Wealth from a discovery decays too quickly to be useful for subsequent detections, leading to a noticeable drop in Power (e.g., Llama+Inv drops from 0.787 at $s = 1.05$ to 0.718 at $s = 2.0$).

- **Recommendation:** We utilize $s = 1.2$ as a standard configuration. It captures the benefits of long memory (high power) while avoiding the potential instability of extremely low decay rates (like $s = 1.0$) in practical, non-stationary deployments.

## D. Experimental Infrastructure & Runtime Efficiency

### D.1. Hardware Specifications

All experiments, including watermark generation and streaming detections were conducted under the following environment:

- **CPU:** Intel(R) Core(TM) i9-14900HX.

- **GPU:** NVIDIA GeForce RTX 4060 Laptop GPU.

- **RAM:** 8 GB.

- **Environment:** Python 3.8.19, PyTorch 2.4.1 (CUDA 12.1).

### D.2. Runtime Analysis

A primary motivation for our online framework is the necessity of low-latency detection in high-throughput API monitoring. While watermark generation time depends on the LLM architecture, the detection phase must be nearly instantaneous.

We benchmarked the processing time per document ($N = 400$ tokens) across all six model-watermark configurations. The average results are presented in Table 16.

*Table 16.* **Runtime Comparison (per document).** Time measured in milliseconds (ms) on an i9-14900HX workstation. The LORD update step introduces negligible overhead ($< 0.001$ ms).

| Component | Naive-GoF (Offline) | LORD-GoF (Online) | Overhead |
|---|---|---|---|
| Pivotal Statistic Extraction | 0.0036 ms | 0.0036 ms | 0.00 ms |
| GoF Calculation (e.g., And) | 0.0124 ms | 0.0124 ms | 0.00 ms |
| Threshold Update (LORD) | N/A | 0.0003 ms | +0.0003 ms |
| **Total Time per Doc** | **0.0160 ms** | **0.0163 ms** | **+1.9%** |

## E. Comparison of Online FDR Procedures

In the main text we adopt **LORD** (Javanmard & Montanari, 2018) as the online FDR engine of LORD-GoF. Several alternatives are also natural candidates: the refined **LORD++** (Ramdas et al., 2017), which uses an asymmetric reward schedule to gain extra power, the adaptive **SAFFRON** (Ramdas et al., 2018), the discarding variant **ADDIS** (Tian & Ramdas, 2019), and e-value-based procedures **e-LORD / e-SAFFRON** (Zhang et al., 2025) which are valid under arbitrary dependence. To make a thorough comparison, we re-run the full sparse-mixed-stream protocol from the main text (stream length $M = 1{,}000$, $\pi = 0.05$, $\rho = 0.7$, $\tau = 0.7$) for *all eight* GoF statistics and *all three* models / *both* watermarks, and report the across-statistic average FDR and Power in Table 17. For the e-value procedures we convert each document p-value $p_t$ into an e-value via the standard calibrator $e_t = \kappa p_t^{\kappa-1} = \kappa/p_t^{1-\kappa}$ with $\kappa \in (0, 1)$, which yields a conditionally valid e-value (it integrates to 1 under a uniform null); we use the default $\kappa = 0.5$, i.e. $e_t = 0.5/\sqrt{p_t}$.

*Table 17.* **Comparison of online FDR procedures used inside LORD-GoF**, averaged over all eight GoF statistics. Configuration: $\alpha = 0.05$, $M = 1000$, $\pi = 0.05$, $\rho = 0.7$, $\tau = 0.7$. Values are reported as **Avg FDR / Avg Power**.

| Model | Watermark | LORD FDR | LORD Power | LORD++ FDR | LORD++ Power | SAFFRON FDR | SAFFRON Power | ADDIS FDR | ADDIS Power | e-LORD FDR | e-LORD Power | e-SAFFRON FDR | e-SAFFRON Power |
|---|---|---|---|---|---|---|---|---|---|---|---|---|---|
| OPT-1.3B | Gumbel | 0.0602 | 0.9303 | 0.0663 | 0.9303 | 0.0615 | 0.9401 | 0.0465 | 0.9313 | 0.0000 | 0.4677 | 0.0000 | 0.4106 |
|  | Inverse | 0.0441 | 0.9026 | 0.0464 | 0.9044 | 0.0358 | 0.9214 | 0.0210 | 0.9062 | 0.0000 | 0.2869 | 0.0000 | 0.2439 |
| Qwen-2.5-3B | Gumbel | 0.0624 | 0.9079 | 0.0659 | 0.9088 | 0.0618 | 0.9110 | 0.0456 | 0.9104 | 0.0000 | 0.6313 | 0.0000 | 0.4068 |
|  | Inverse | 0.0463 | 0.8428 | 0.0516 | 0.8445 | 0.0400 | 0.8558 | 0.0280 | 0.8465 | 0.0000 | 0.1694 | 0.0000 | 0.0798 |
| Sheared-LLaMA-2.7B | Gumbel | 0.0623 | 0.9083 | 0.0658 | 0.9092 | 0.0607 | 0.9125 | 0.0449 | 0.9092 | 0.0000 | 0.6256 | 0.0000 | 0.4317 |
|  | Inverse | 0.0386 | 0.7874 | 0.0452 | 0.7917 | 0.0375 | 0.8097 | 0.0231 | 0.7962 | 0.0000 | 0.1321 | 0.0000 | 0.0557 |

## E.1. Analysis and Selection

Several patterns emerge from Table 17. Among p-value-based procedures, **SAFFRON** is typically the most powerful, delivering the highest power in every configuration (e.g., $0.9214$ vs $0.9044$ on OPT+Inverse), while **ADDIS** is the most conservative, reaching the lowest FDR in every configuration (dipping to $\approx 0.02$ on the Inverse-Transform settings while sitting near $0.045$ on Gumbel-Max). The original **LORD** and the refined **LORD++** sit in between and produce very similar FDR/Power profiles across all six configurations; the asymmetric reward of LORD++ provides only a marginal power gain over LORD ($\leq 0.006$ absolute on every configuration) at a slightly higher empirical FDR. Either is a reasonable default; we use LORD because it is the simplest of the four p-value-based procedures and matches the implementation used throughout our experiments.

The e-value-based procedures **e-LORD** and **e-SAFFRON** achieve essentially zero empirical FDR on every configuration, which is consistent with their key theoretical advantage: online FDR control under *arbitrary* dependence of the p-value stream (Zhang et al., 2025). The price for this robustness is power: in the sparse, mixed-stream regime considered here, their average power is substantially lower than that of the p-value-based procedures (often by 0.3–0.7 absolute), reflecting the inherent looseness of the p-to-e calibrator $e = \kappa/p^{1-\kappa}$ ($\kappa = 0.5$). We therefore recommend e-LORD/e-SAFFRON whenever the user has reason to doubt the independence of document-level p-values across the stream, and **LORD** as the default for typical sparse-mixed deployment, where its simplicity and balanced behavior provide a strong default-of-choice for LORD-GoF.

