# OpenReview forum: "LORD-GoF: A Robust Online Detection Approach for LLM Watermarks in Sparse and Mixed Streams"
_ICML.cc/2026/Conference — ICML 2026 regular_

### Official Review · Reviewer_zN5o · 2026-02-28

**Soundness:** 3
**Presentation:** 4
**Significance:** 2
**Originality:** 3
**Overall Recommendation:** 3
**Confidence:** 4

**Summary:**

The authors propose a method to control the Online False Discovery Rate (oFDR) for LLM watermark detection in streaming scenarios where watermarked content is sparse. To address this, the study integrates Goodness-of-Fit (GoF) statistics with the LORD++ (Levels based On Recent Discovery) procedure. Simulation results demonstrate that the proposed method effectively controls the oFDR at the desired level.

**Compliance With Llm Reviewing Policy:**

Affirmed.

**Final Justification:**

The paper requires additional refinement to reach a higher level of impact.

**Key Questions For Authors:**

The toy example provided to illustrate the failure of offline methods is misleading and reveals a fundamental misunderstanding of statistical hypothesis testing. The authors state: “A static detector controlling the False Discovery Rate at $\alpha = 0.05$ will... flag $100 \times 0.05 = 5$ null texts.” This description describes a detector controlling the Type I error rate (marginal significance level) at 0.05, not the FDR. FDR control is inherently a multiple testing problem, and standard offline procedures (such as the Benjamini-Hochberg procedure) are specifically designed to adapt thresholds to maintain the FDR below a target level, preventing the exact scenario the authors describe. By conflating Type I error control with FDR control, the authors construct a misleading example.

**Limitations:**

The motivation for the study is not sufficiently grounded in reality.

**Strengths And Weaknesses:**

The paper is well-organized and clearly written. The authors provide a comprehensive introduction to the background and the methodology, ensuring the manuscript is accessible and easy to follow. However, I have following concerns about the proposed method.

1.  Insufficient Motivation and Practical Relevance. The primary concern regarding this work is the validity of the problem setting. The motivation for the study is not sufficiently grounded in reality. It is difficult to envision a practical real-world scenario involving a continuous stream of documents that requires instantaneous, sequential watermark detection at every step. The authors need to provide a more concrete and detailed use case to substantiate the necessity of such a streaming setting; otherwise, the problem appears artificial.

2.  Limited Technical Novelty. The proposed method appears to be a straightforward concatenation ("1 + 1") of GoF statistics and the LORD++ procedure. Consequently, the application scenario feels somewhat contrived, as if it were constructed primarily to justify the combination of these two specific existing techniques rather than to solve an organic problem.

3.  Lack of Comprehensive Exploration. Even assuming the validity of the application scenario, the exploration of the solution space is inadequate. There are numerous existing methods for watermark detection beyond GoF statistics, just as there are various alternative approaches for oFDR control. If the authors aim to investigate the intersection of these two domains, a broader comparative analysis involving different detection metrics and control algorithms is essential to validate their design choices.

---

> ### Author Rebuttal · Authors · 2026-03-31
>
> Thank you for the careful review. We appreciate the positive comments on clarity and organization and address the main concerns below.
>
> **Q1. Motivation and practical value of the streaming setting**
>
> We agree that watermark detection includes both **batch** and **online** use. In practice, it is not rare for a user to submit **a single document** to a detector for checking; this is a natural **streaming** case. Batch use also clearly exists—for example, a teacher may collect many assignments and check them together. Existing tools support both modes (e.g., GPTZero supports up to 250 files; Originality.ai supports bulk scans [1,2], while Copyleaks and Stream illustrate real-time use [3,4]). Our point is not that all detection must be streaming, but that **streaming**—documents arriving one by one, unknown total length, and one-shot decisions—is a realistic and under-studied setting. We will revise the paper to make this scope clearer.
>
> **Q2. Novelty and contribution**
>
> We agree that we do not introduce a new GoF statistic or online testing rule. Our contribution is to address a practical deployment mismatch, since real watermark detection is often **sparse, mixed, and sequential** while prior evaluation is mostly **offline** with fixed thresholds, using a **simple, principled, and deployable** combination of GoF detection and online FDR control.
>
> **Q3. Breadth of exploration and comparison**
>
> We agree that broader exploration is important. The paper already studies **multiple models, watermark schemes, sparsity levels, edit ratios, temperatures, and GoF statistics**. We also added experiments on **Google SynthID** [5] and compared raw **GoF** with **LORD-GoF** under the same sparse mixed-stream setting.
>
> We do not include separate **KGW / green-red list** experiments because its binary pivotal statistic makes detection essentially a **count-based test**; as discussed in [6], GoF tests here largely reduce to the original KGW detector.
>
> The main result is consistent: under **SynthID**, raw GoF has very high power but **very high FDR**, while **LORD-GoF sharply lowers FDR** and keeps strong power for several statistics.
>
> **OPT-1.3B + SynthID [5]:**
> | GoF statistic | GoF FDR | GoF Power | LORD-GoF FDR | LORD-GoF Power |
> |---|---:|---:|---:|---:|
> | Anderson | 0.533 | 1.000 | 0.070 | 0.979 |
> | Chi-squared | 0.494 | 1.000 | 0.055 | 0.957 |
> | Cramer | 0.523 | 1.000 | 0.072 | 0.972 |
> | Greenwood | 0.534 | 0.987 | 0.051 | 0.652 |
> | Kolmogorov | 0.520 | 1.000 | 0.063 | 0.935 |
> | Kuiper | 0.495 | 1.000 | 0.037 | 0.853 |
> | Rao | 0.554 | 0.994 | 0.077 | 0.659 |
> | Watson | 0.514 | 1.000 | 0.060 | 0.766 |
>
> **Q4. Toy example: Type I error vs. FDR**
>
> We agree that the original toy example used the terms imprecisely: it described control of the *marginal Type I error rate* at $\alpha=0.05$, not control of **FDR**. We will fix this. We also agree that in a **closed batch** setting, where all p-values are known in advance, standard offline methods such as **Benjamini–Hochberg procedure** are the right tool; our earlier wording was too broad for that case.
>
> Our target setting is **online streaming**: documents arrive one by one, and each $D_t$ must be decided *when it arrives*, without future information. In this case, the rule $p_t \le \alpha$ is just repeated single-test control. In the toy example (100 nulls + 5 alternatives), this gives about 5 false positives. Even if all alternatives are correctly identified,   the $\mathrm{FDR}\approx 0.5$. We will revise the example to clearly separate marginal Type I error, offline batch FDR, and valid **online FDR control** for streams.
>
> ## **References**
> [1] GPTZero. *Frequently Asked Questions*. Official website. States that users can run detection on “a batch of up to 250 files.” [https://gptzero.me/faq](https://gptzero.me/faq)
>
> [2] Originality.ai. *Bulk Scan*. Official help page. Describes batch processing of multiple texts or URLs via CSV upload. [https://help.originality.ai/en/article/bulk-scanner-5uqkpb/](https://help.originality.ai/en/article/bulk-scanner-5uqkpb/)
>
> [3] Copyleaks. *Copyleaks Google Docs Add-on*; *Write Smarter, Verify Faster: Copyleaks Launches Google Docs Add-on for Real-Time AI and Plagiarism Detection*. Official product/blog pages. [https://copyleaks.com/blog/copyleaks-launches-google-docs-add-on](https://copyleaks.com/blog/copyleaks-launches-google-docs-add-on)
>
> [4] Stream. *AI Content Moderation API*; *Overview of AI Moderation*. Official product/course pages. [https://getstream.io/resources/projects/moderation-course/moderator/overview/](https://getstream.io/resources/projects/moderation-course/moderator/overview/)
>
> [5] Dathathri et al. *Scalable watermarking for identifying large language model outputs*. Nature, 2024.
>
> [6] He, W., Li, X., Shang, T., Shen, L., Su, W. J., and Long, Q. *On the Empirical Power of Goodness-of-Fit Tests in Watermark Detection.* NeurIPS, 2025.

---

> > ### Author Rebuttal · Reviewer_zN5o · 2026-04-02
> >
> > Dear Authors,
> >
> > Thank you for the detailed rebuttal.
> >
> > I appreciate the clarifications on the streaming setting and the correction regarding Type I error vs. FDR; these partially address my concerns. The additional experiments also improve the empirical support.
> >
> > However, compared to "On the Empirical Power of Goodness-of-Fit Tests in Watermark Detection", this work is still less convincing in terms of both technical depth and overall completeness. I believe the paper would benefit from further strengthening in these aspects.
> >
> > Overall, I will raise my score from 2 to 3, but I still believe the paper requires additional refinement to reach a higher level of impact.

---

> > > ### Author Response · Authors · 2026-04-06
> > >
> > > Thank you very much for the thoughtful follow-up and for raising your score.
> > >
> > > We appreciate your comment and agree that the paper can be further strengthened. In the revision, we will continue refining the paper in terms of presentation, positioning, and overall completeness.
> > >
> > > More specifically, we will make clearer that our focus is a deployment issue that is not fully addressed in current watermark detection work. In many existing studies, including but not limited to "On the Empirical Power of Goodness-of-Fit Tests in Watermark Detection," evaluation is still mainly based on offline, roughly balanced samples with a fixed significance threshold, while our paper focuses on sequential and highly imbalanced deployment settings.
> > >
> > > Thank you again for your careful reading and constructive feedback.

---

### Official Review · Reviewer_1rUF · 2026-03-11

**Soundness:** 3
**Presentation:** 3
**Significance:** 2
**Originality:** 3
**Overall Recommendation:** 3
**Confidence:** 4

**Summary:**

This paper studies the problem of detecting LLM-generated text in online streaming settings using watermark signals. While most existing watermark detection methods operate offline, the authors argue that real-world deployment often involves mixed streams of human- and machine-generated content, requiring sequential decisions with controlled false discoveries. The paper proposes LORD-GoF, which combines a goodness-of-fit (GoF) statistic for watermark detection with the LORD procedure for online multiple testing. Experiments on several LLMs and watermarking schemes show that the method maintains comparable detection power to offline baselines while controlling the online false discovery rate (oFDR).

**Compliance With Llm Reviewing Policy:**

Affirmed.

**Final Justification:**

The rebuttal is helpful and improves the paper, particularly through additional robustness experiments under realistic text transformations.

Nevertheless, my primary concern about methodological novelty is not fully addressed. The proposed approach mainly integrates existing statistical tools (GoF-based detection and LORD for online FDR control), and the rebuttal does not sufficiently clarify what fundamentally new methodological insight is introduced.

As a result, while the empirical section is strengthened, my overall evaluation of originality and contribution remains unchanged. I therefore maintain my original score.

**Key Questions For Authors:**

Please see weaknesses.

**Limitations:**

yes

**Strengths And Weaknesses:**

**Strengths**

- **Well-motivated problem setting.**
The paper highlights a practical limitation of existing watermark detection approaches, which typically assume offline analysis, while real-world systems may operate on continuous streams of mixed human- and machine-generated text.

- **Clear statistical formulation and theoretical guarantees.**
The problem is formulated as an online hypothesis testing task. The combination of a GoF statistic with the LORD procedure provides a clear framework grounded in the literature on online multiple testing. Moreover, the method provides theoretical guarantees on controlling the online false discovery rate (oFDR), which offers a principled alternative to heuristic thresholding approaches.

- **Empirical validation.**
Experiments on several LLMs and watermarking schemes show that the proposed method can maintain detection power while controlling oFDR in simulated streaming scenarios.

**Weaknesses**

- **Limited methodological novelty.**
The proposed framework mainly combines existing GoF-based watermark detection statistics with the LORD procedure from the online multiple testing literature. As a result, the methodological contribution appears relatively incremental.

- **Limited insight beyond the integration of existing techniques.**
While the statistical formulation is sound, the paper provides limited new insight into watermark detection itself, and the main contribution largely lies in applying existing statistical tools to this setting.

- **Limited evaluation scope.**
The experiments focus on a small number of watermarking schemes and simulated streaming settings. It remains unclear how well the proposed framework generalizes to other watermark designs or more realistic deployment scenarios.

- **Robustness to realistic text transformations is not examined.**
The evaluation mainly considers clean generation settings. It would be valuable to analyze performance under common text transformations such as paraphrasing or editing, which may weaken watermark signals in practice.

---

> ### Author Rebuttal · Authors · 2026-03-31
>
> Thank you for the thoughtful review and constructive suggestions. We respond to the main concerns below.
>
> **Q1. Methodological novelty**
>
> We agree that we do not introduce a new GoF statistic or online testing rule. Our contribution is to address a practical deployment mismatch, since real watermark detection is often **sparse, mixed, and sequential** while prior evaluation is mostly **offline** with fixed thresholds, using a **simple, principled, and deployable** combination of GoF detection and online FDR control.
>
> **Q2. Additional insight beyond combining existing tools**
>
> Thank you for this question. Beyond simply combining two existing components, our main insight is that **online deployment changes the optimization target of watermark detection**. In sparse streams, the main problem is not weak document-level evidence, but the accumulation of false positives under repeated fixed-threshold testing. A strong detector paired with **dynamically allocated testing levels** makes the whole pipeline more effective: LORD++ adapts the rejection threshold over time, so the procedure can **control oFDR while preserving as much power as possible**. We also find this combination to be practically robust: in our experiments, the feared **alpha-death** behavior is rare, suggesting that LORD++ is well matched to realistic watermark streams. We will revise the paper to make this deployment-level insight clearer.
>
> **Q3. Breadth of experiments and generalization to other watermark schemes**
>
> Thank you for this important comment. We agree that broader exploration is important. Our paper already studies multiple models, watermark schemes, sparsity levels, edit ratios, temperatures, and GoF statistics. We do not include separate **KGW / green-red list** experiments because its binary pivotal statistic makes detection essentially a **count-based test**; as discussed in [3], GoF tests here largely reduce to the original KGW detector. To further strengthen this point, we now add experiments on **SynthID** beyond the OPT/Qwen settings in the paper. The results are summarized below.
>
> **OPT-1.3B + SynthID [1]:**
>
> | GoF statistic | GoF FDR | GoF Power | LORD-GoF FDR | LORD-GoF Power |
> |---|---:|---:|---:|---:|
> | Anderson | 0.533 | 1.000 | 0.070 | 0.979 |
> | Chi-squared | 0.494 | 1.000 | 0.055 | 0.957 |
> | Cramer | 0.523 | 1.000 | 0.072 | 0.972 |
> | Greenwood | 0.534 | 0.987 | 0.051 | 0.652 |
> | Kolmogorov | 0.520 | 1.000 | 0.063 | 0.935 |
> | Kuiper | 0.495 | 1.000 | 0.037 | 0.853 |
> | Rao | 0.554 | 0.994 | 0.077 | 0.659 |
> | Watson | 0.514 | 1.000 | 0.060 | 0.766 |
>
> **Qwen-2.5-3B + SynthID [1]:**
>
> | GoF statistic | GoF FDR | GoF Power | LORD-GoF FDR | LORD-GoF Power |
> |---|---:|---:|---:|---:|
> | Anderson | 0.544 | 0.957 | 0.056 | 0.874 |
> | Chi-squared | 0.509 | 0.943 | 0.053 | 0.754 |
> | Cramer | 0.536 | 0.951 | 0.059 | 0.831 |
> | Greenwood | 0.530 | 1.000 | 0.050 | 0.928 |
> | Kolmogorov | 0.531 | 0.957 | 0.053 | 0.832 |
> | Kuiper | 0.510 | 0.943 | 0.031 | 0.769 |
> | Rao | 0.929 | 0.090 | 0.000 | 0.055 |
> | Watson | 0.564 | 0.817 | 0.054 | 0.477 |
>
> These results further support that the benefit is not tied to a single model or watermark scheme, but comes from combining document-level detection evidence with online FDR control.
>
> **Q4. Robustness under realistic text transformations**
>
> Thank you for pointing this out. Our $\rho$-based setting already simulates three post-editing operations, **deletion, insertion, and substitution**, rather than a purely abstract mixture. To further strengthen robustness evaluation, we additionally include **round-trip translation** and **DIPPER-based paraphrasing** attacks [2]. The results show the same overall trend: stronger edits reduce absolute power, but the online control framework remains substantially more stable than naive fixed-threshold deployment. Compared with the paper’s main settings, oFDR remains well-controlled, while Power decreases moderately  because strong edits weaken the watermark signal used in hash-based verification.
>
> **OPT-1.3B:**
> | Edit setting | Watermark | Avg. FDR | Avg. Power |
> |---|---|---:|---:|
> | Round-trip translation (EN to FR to EN) | Gumbel | 0.03 | 0.76 |
> | Round-trip translation (EN to FR to EN) | Inverse | 0.03 | 0.78 |
> | DIPPER paraphrase | Gumbel | 0.02 | 0.65 |
> | DIPPER paraphrase | Inverse | 0.03 | 0.70 |
>
> **Sheared-LLaMA-2.7B:**
> | Edit setting | Watermark | Avg. FDR | Avg. Power |
> |---|---|---:|---:|
> | Round-trip translation (EN to FR to EN) | Gumbel | 0.03 | 0.77 |
> | Round-trip translation (EN to FR to EN) | Inverse | 0.04 | 0.79 |
> | DIPPER paraphrase | Gumbel | 0.02 | 0.65 |
> | DIPPER paraphrase | Inverse | 0.02 | 0.71 |
>
> **References**
>
> [1] Dathathri et al. Scalable watermarking for identifying large language model outputs. Nature, 2024.
>
> [2] Krishna et al. Paraphrasing evades detectors of AI-generated text, but retrieval is an effective defense. NeurIPS, 2023.
>
> [3] He et al. On the Empirical Power of Goodness-of-Fit Tests in Watermark Detection. NeurIPS, 2025.

---

> > ### Author Rebuttal · Reviewer_1rUF · 2026-04-04
> >
> > The authors provide a thoughtful rebuttal and additional experiments addressing several of my concerns. In particular, the added robustness evaluation under realistic text transformations (e.g., paraphrasing and round-trip translation) is valuable and strengthens the empirical section.
> >
> > However, my main concern regarding methodological novelty remains largely unresolved. The proposed framework primarily combines existing GoF-based watermark detection with established online FDR control (LORD), and the rebuttal does not sufficiently clarify what fundamentally new insight or technique is introduced beyond this integration.
> >
> > While the authors argue that online deployment changes the optimization objective, this perspective is not fully developed into a novel methodological contribution, but rather reflects an application of known statistical tools to a new setting.
> >
> > Overall, the rebuttal improves the experimental validation, but does not substantially change my assessment of the paper’s originality and core contribution.

---

> > > ### Author Response · Authors · 2026-04-06
> > >
> > > Thank you very much for the thoughtful follow-up. We appreciate your recognition that the additional robustness experiments strengthen the empirical section.
> > >
> > > We also understand your remaining concern regarding methodological novelty. We agree that this paper does not introduce a brand-new standalone method. Instead, our main contribution is to address a practical real-world deployment problem in a clean and principled way.
> > >
> > > We will revise the paper to make this positioning clearer. Thank you again for your careful and constructive feedback.

---

### Official Review · Reviewer_19Ke · 2026-03-12

**Soundness:** 3
**Presentation:** 3
**Significance:** 2
**Originality:** 2
**Overall Recommendation:** 4
**Confidence:** 2

**Summary:**

This paper proposes LORD-GoF, a streaming detection framework for LLM watermarks that combines (i) document-level p-values obtained from Goodness-of-Fit (GoF) tests applied to token-level pivotal statistics and (ii) online false discovery rate (oFDR) control via the LORD++ procedure. The authors prove oFDR control under standard super-uniformity and independence assumptions, and empirically demonstrate that LORD-GoF maintains oFDR near the target level while achieving high power in sparse, mixed streams, outperforming static thresholding baselines on multiple models and watermark schemes.

**Compliance With Llm Reviewing Policy:**

Affirmed.

**Final Justification:**

Because my previous concerns are fully resolved, I raise my score from 3 to 4.

**Key Questions For Authors:**

Please see the Strengths and Weaknesses section. And some additional questions are as follows:

1. How sensitive are results to LORD++ hyperparameters (W0, γ schedule)? A systematic ablation (including the “alpha death” phenomenon) would clarify robustness.

2. Do you have experiments with real human edits/paraphrase/translation or attack-style edits beyond a synthetic mixture parameter ρ? If so, how do the results compare to the simulated setting?

**Limitations:**

yes

**Strengths And Weaknesses:**

Strengths:

1.Timely and Deployment-Oriented Problem Formulation. The shift from offline, balanced evaluation protocols to sparse, streaming settings is well motivated. The paper correctly identifies that fixed significance thresholds are ill-suited to guaranteeing online FDR control in sequential settings, and that this mismatch can lead to inflated false discoveries in practice. This is an important and practically relevant observation with direct implications for real-world watermark deployment.

2.Clean Integration of Principled Statistical Components. The framework combines two well-established components: (i) goodness-of-fit-based p-value construction for watermark detection, and (ii) online FDR control via LORD++. The resulting pipeline is modular and conceptually straightforward—a notable strength from both an engineering and a deployment perspective. The theoretical guarantee (Theorem 4.1) follows naturally under the stated assumptions, and the accompanying argumentation is logically coherent throughout.

3.Systematic and Multi-Dimensional Experimental Study. The empirical evaluation explores several factors of practical relevance, including global sparsity (π), local watermark density under editing (ρ), generation temperature (τ), as well as multiple language models and watermarking schemes. The comparisons between static thresholds and LORD-based dynamic thresholds effectively illustrate the risk of FDR inflation in sparse streams, providing compelling evidence for the proposed approach.

Weaknesses:

1.Limited Methodological Novelty. The core contribution largely consists of combining existing GoF-based watermark detectors with the LORD++ algorithm, without introducing new statistical principles tailored to watermark structure. Clarifying the watermark-specific technical challenges that necessitate nontrivial adaptation would better justify the contribution beyond straightforward integration.

2.Under Examined Theoretical Assumptions. The online FDR guarantee relies on super-uniformity of p-values and predictability of testing levels—standard conditions whose validity under realistic watermark settings (e.g., token-level dependence, correlated document streams, post-editing artifacts) is neither discussed nor empirically verified.

3.Narrow Baseline Comparison. Only a fixed αα-level detector serves as baseline. Including established online FDR methods (e.g., SAFFRON, ADDIS, e-value-based procedures) would clarify whether the observed gains are attributable to LORD++ specifically or to online FDR control in general.

4. Minor Presentation Issues. Hyperparameter inconsistencies (e.g., W0=0.1α in the text versus W0=0.01 in experiments) and an underspecified p-value calibration procedure (exact, asymptotic, or Monte Carlo) should be resolved to ensure reproducibility.

---

> ### Author Rebuttal · Authors · 2026-03-31
>
> Thank you for careful review and suggestions. We respond to the main concerns below.
>
> **Q1. Methodological novelty**
>
> We agree that we do not introduce a new GoF statistic or online testing rule. Our contribution is to address a practical deployment mismatch, since real watermark detection is often **sparse, mixed, and sequential** while prior evaluation is mostly **offline** with fixed thresholds, using a **simple, principled, and deployable** combination of GoF detection and online FDR control.
>
> **Q2. Theoretical assumptions**
>
> Thank you for highlighting this issue. At the **token level**, although text is autoregressive, watermark detection is built on **pivotal statistics**: under $H_0$, the pseudorandomness generated by hash-based construction with the secret key ensures that token-level statistics are i.i.d. from a known null distribution [1,2].
>
> At the **document level**, we agree that full independence is restrictive. Recent work suggests p-value-based LORD/SAFFRON can extend to weaker **PRDS dependence** [3], while e-value-based procedures can handle **arbitrary dependence** [4]. We will handle this limitation in future work.
>
> **Q3. Narrow baseline comparison**
>
> We agree fixed-threshold baselines alone are insufficient. We have therefore added stronger online FDR baselines ; due to the rebuttal length limit, we kindly refer the reviewer to our response to **Reviewer QAeL(Q2)** for the added experiments.
>
> **Q4. Minor presentation and reproducibility**
>
> Thank you for catching these issues. Our code uses $\alpha=0.05$ and $W_0=0.01$, so the statement $W_0=0.1\alpha$ is incorrect and will be fixed. Document-level p-values are calibrated by Monte Carlo. Code has been submitted as supplementary material and will be released upon acceptance.
>
> **Q5. Sensitivity to Hyperparameters**
>
> Thank you for the question. We agree that sensitivity to hyperparameters deserves clearer discussion. In addition to the ablations already in **Appendix A.3**, we summarize two key results.
>
> First, LORD-GoF is stable to standard hyperparameters. Smaller $W_0$ delays early detections; larger decay exponent $s$ focuses more on recent discoveries.
>
> **Sensitivity to initial wealth $W_0$** (fixed $s=1.2$)
> FDR / Power
> | $W_0$ | OPT+Gum | OPT+Inv | Qwen+Gum | Qwen+Inv |
> |---|---|---|---|---|
> | 0.0005 | 0.068/0.816 | 0.046/0.916 | 0.063/0.853 | 0.047/0.823 |
> | 0.0050 | 0.052/0.819 | 0.041/0.925 | 0.055/0.850 | 0.043/0.823 |
> | 0.0100 | 0.048/0.818 | 0.036/0.923 | 0.050/0.848 | 0.036/0.816 |
> | 0.0250 | 0.031/0.803 | 0.024/0.910 | 0.036/0.842 | 0.023/0.800 |
>
> **Sensitivity to decay exponent $s$** (fixed $W_0=0.005$)
> FDR / Power
> | $s$ | OPT+Gum | OPT+Inv | Qwen+Gum | Qwen+Inv |
> |---|---|---|---|---|
> | 1.05 | 0.054/0.827 | 0.042/0.931 | 0.055/0.855 | 0.044/0.828 |
> | 1.20 | 0.052/0.819 | 0.041/0.925 | 0.055/0.850 | 0.043/0.823 |
> | 1.60 | 0.056/0.797 | 0.044/0.908 | 0.059/0.845 | 0.043/0.793 |
> | 2.00 | 0.058/0.766 | 0.040/0.892 | 0.062/0.837 | 0.046/0.762 |
>
> Second, to directly address **“alpha death”** , we ran a stress test on OPT-1.3B: a long null-only prefix followed by a fixed watermarked suffix. These results suggest that the alpha-death phenomenon is limited in our setting: even after long null-only prefixes, non-trivial wealth remains and power stays near 1 once watermarked signals appear.
>
> **Alpha-death stress test (OPT-1.3B, $\alpha=0.05$, $W_0=0.01$, $s=1.2$)**
> Power after signal appears / Remaining wealth after prefix
> | Null prefix length | Gumbel Power | Inverse Power | Gumbel Wealth | Inverse Wealth |
> |--------------------|--------------|---------------|---------------|----------------|
> | $10^2$           | 0.99965      | 0.99975       | 0.0063        | 0.0056         |
> | $10^3$           | 0.99950      | 0.99995       | 0.0065        | 0.0051         |
> | $10^4$           | 0.99955      | 0.99970       | 0.0120        | 0.0135         |
>
> **Q6. Realistic edits beyond the synthetic $\rho$-based setting**
>
> Thank you for pointing this out. Our $\rho$-based setting is not purely abstract; it simulates **deletion, insertion, and substitution**. We also considered round-trip translation and DIPPER paraphrasing [5]. Due to space limits, please see our response to **Reviewer 1rUF(Q4)** for those results.
>
> **References**
>
> [1] Li et al. *A Statistical Framework of Watermarks for Large Language Models: Pivot, Detection Efficiency and Optimal Rules.* The Annals of Statistics, 2025.
>
> [2] Li et al. *Robust Detection of Watermarks for Large Language Models Under Human Edits.* Journal of the Royal Statistical Society: Series B (Statistical Methodology), 2025.
>
> [3] Fisher, A. *Online False Discovery Rate Control for LORD & SAFFRON Under Positive, Local Dependence.* Biometrical Journal, 2024.
>
> [4] Zhang et al. *e-GAI: e-value-based Generalized $\alpha$-Investing for Online False Discovery Rate Control.* ICML, 2025.
>
> [5] Krishna et al. *Paraphrasing evades detectors of AI-generated text, but retrieval is an effective defense.* NeurIPS, 2023.

---

> > ### Author Rebuttal · Reviewer_19Ke · 2026-04-05
> >
> > My previous concerns are fully resolved. And I decide to raise my score from 3 to 4.

---

> > > ### Author Response · Authors · 2026-04-06
> > >
> > > Thank you very much for your encouraging follow-up and for raising your score from 3 to 4. We are very glad that our rebuttal addressed your concerns. We sincerely appreciate your careful reading and constructive feedback, and we will incorporate the clarifications and additional experimental results into the revised version.

---

### Official Review · Reviewer_QAeL · 2026-03-13

**Soundness:** 3
**Presentation:** 3
**Significance:** 3
**Originality:** 3
**Overall Recommendation:** 5
**Confidence:** 3

**Summary:**

This paper studies online detection of LLM watermarks in sparse, mixed document streams, arguing that standard offline detectors with a fixed threshold are statistically mismatched to realistic deployments because they do not control the online false discovery rate (oFDR) when positives are rare. The proposed method, LORD-GOF, combines document-level goodness-of-fit (GoF) p-values derived from token-level pivotal statistics with the LORD++ online multiple testing procedure, and proves oFDR control under assumptions stated in the appendix. Empirically, on C4 using three open-source models (OPT-1.3B, Sheared-LLaMA-2.7B, Qwen-2.5-3B) and two distortion-free watermarking schemes (Gumbel-Max and Inverse Transform), the method generally keeps FDR near the target while preserving strong power, especially compared with naive fixed-threshold baselines in sparse streams.

**Compliance With Llm Reviewing Policy:**

Affirmed.

**Final Justification:**

My concerns have been addressed.

**Key Questions For Authors:**

Please see above.

**Limitations:**

Please see above.

**Strengths And Weaknesses:**

Strengths

1. The paper addresses a realistic and practically relevant deployment scenario: online detection in sparse, mixed streams rather than offline balanced evaluation.

2. The method is simple, statistically grounded, and easy to understand: GoF-based p-values plus online FDR control via LORD++.

3. Experiments convincingly show that naive fixed-threshold baselines can fail badly in sparse streams, while the proposed approach usually maintains much better FDR control.

Weaknesses:

1. The guarantees rely on assumptions such as independence and well-calibrated null p-values, which may not hold in realistic correlated document streams.

2. Much of the empirical advantage comes from outperforming a naive fixed-threshold baseline, which is not a very strong competitor.

3. Some reported settings still exceed the target FDR, suggesting that performance is sensitive to the choice of GoF statistic.

---

> ### Author Rebuttal · Authors · 2026-03-31
>
> Thank you for the thoughtful and constructive review. We appreciate your positive assessment and address the three concerns below.
>
> **Q1. Assumptions and correlated streams**
>
> We agree that our theorem is currently proved under a restrictive assumption: under $H_{0,t}$, $P_t \perp \mathcal{F}_{t-1}$. This yields the conditional super-uniformity required by LORD++ [1,2], and we will state this limitation more clearly.
>
> However, we note recent results suggest that for **p-value-based** LORD/SAFFRON, online FDR control can extend to a weaker **positive local dependence / PRDS** condition [7], weakening our assumption from independence to PRDS. More broadly, **e-value-based** procedures offer a promising route to more general **arbitrary dependence** under conditional e-value validity [3]. We will clarify both points in the revision.
>
> **Q2. Stronger baselines beyond fixed thresholding**
>
> We agree that comparing only with a fixed-threshold detector is insufficient. That baseline was included mainly to isolate the sparse-stream deployment mismatch studied in this paper. Note that in Appendix A.5, we compared LORD, LORD++, SAFFRON, and ADDIS with a subset of GoF statistics. In the revision, we will make this comparison more prominent by reporting results on all three models, both watermark schemes, and all eight GoF statistics. **All results here use temperature 0.7.** We also include **e-LORD** and **e-SAFFRON**, obtained by converting the same p-values into e-values via
> $$
> e(p)=\frac{\kappa}{p^{\,1-\kappa}}, \qquad \kappa\in(0,1),
> $$
> with default
>  $\(\kappa=0.5\)$,
> | Scenario | ADDIS (FDR/Power) | LORD (FDR/Power) | LORD++ (FDR/Power) | SAFFRON (FDR/Power) | e-LORD (FDR/Power) | e-SAFFRON (FDR/Power) |
> |---|---|---|---|---|---|---|
> | OPT-1.3B + Gumbel | 0.0465 / 0.9313 | 0.0602 / 0.9303 | 0.0663 / 0.9303 | 0.0615 / 0.9401 | 0.0000 / 0.4677 | 0.0000 / 0.4106 |
> | OPT-1.3B + Inverse | 0.0210 / 0.9062 | 0.0441 / 0.9026 | 0.0464 / 0.9044 | 0.0358 / 0.9214 | 0.0000 / 0.2869 | 0.0000 / 0.2439 |
> | Qwen-2.5-3B + Gumbel | 0.0456 / 0.9104 | 0.0624 / 0.9079 | 0.0659 / 0.9088 | 0.0618 / 0.9110 | 0.0000 / 0.6313 | 0.0000 / 0.4068 |
> | Qwen-2.5-3B + Inverse | 0.0280 / 0.8465 | 0.0463 / 0.8428 | 0.0516 / 0.8445 | 0.0400 / 0.8558 | 0.0000 / 0.1694 | 0.0000 / 0.0798 |
> | Sheared-LLaMA + Gumbel | 0.0449 / 0.9092 | 0.0623 / 0.9083 | 0.0658 / 0.9092 | 0.0607 / 0.9125 | 0.0000 / 0.6256 | 0.0000 / 0.4317 |
> | Sheared-LLaMA + Inverse | 0.0231 / 0.7962 | 0.0386 / 0.7874 | 0.0452 / 0.7917 | 0.0375 / 0.8097 | 0.0000 / 0.1321 | 0.0000 / 0.0557 |
>
> Overall, the improvement comes from **mainstream online FDR control**, not from beating a weak fixed-threshold baseline. Among p-value baselines, **SAFFRON** is usually the most aggressive and often has the highest power, while **ADDIS** is typically the most conservative in FDR. **LORD++ is not uniformly best, but remains broadly competitive and simpler than SAFFRON/ADDIS, which require extra tuning parameter.** We also report **e-LORD** and **e-SAFFRON**: their main advantage is valid online FDR control under **arbitrary dependence**, but this robustness comes with much lower power. We will revise the paper accordingly and rephrase the expression.
>
> **Q3. Sensitivity to the GoF statistic and occasional FDR exceedance**
>
> We agree that detection performance depends on the chosen GoF statistic. The key issue is null calibration: at **low temperatures**, repetitive outputs induce local dependence, which can make some GoF p-values slightly anti-conservative and thus cause mild FDR inflation. At **higher temperatures**, the null is closer to the i.i.d. ideal and watermarking induces clearer distributional shifts, so most GoF tests achieve better control and power. Under stronger edits, tests less dominated by a few extremes (e.g., Kuiper, Watson, Chi-squared) tend to be more robust [6].
>
>
> **References**
>
> [1] Javanmard, A. and Montanari, A. *Online rules for control of false discovery rate and false discovery exceedance.* Ann. Stat., 2018.
> [2] Ramdas, A., Yang, F., Wainwright, M. J., and Jordan, M. I. *Online control of the false discovery rate with decaying memory.* NeurIPS, 2017.
> [3] Zhang, Y., Wei, Z., Ren, H., and Zou, C. *e-GAI: e-value-based Generalized $\alpha$-Investing for Online False Discovery Rate Control.* ICML, 2025.
> [4] Ramdas, A., Zrnic, T., Wainwright, M. J., and Jordan, M. I. *SAFFRON: an adaptive algorithm for online control of the false discovery rate.* ICML, 2019.
> [5] Tian, J. and Ramdas, A. *ADDIS: an adaptive discarding algorithm for online FDR control with conservative nulls.* NeurIPS, 2019.
> [6] He, W., Li, X., Shang, T., Shen, L., Su, W. J., and Long, Q. *On the Empirical Power of Goodness-of-Fit Tests in Watermark Detection.* NeurIPS, 2025.
> [7]  Fisher, A. *Online False Discovery Rate Control for LORD & SAFFRON Under Positive, Local Dependence.* Biometrical Journal, 2024.

---

> > ### Author Rebuttal · Reviewer_QAeL · 2026-04-01
> >
> > Thank you for your detailed response. My concerns have been addressed, and I have raised my score.

---

> > > ### Author Response · Authors · 2026-04-03
> > >
> > > Thank you very much for reading our rebuttal carefully and for your encouraging feedback. We are glad that our response addressed your concerns, and we sincerely appreciate your support and updated score. We will incorporate the clarifications and additional experimental results into the revised version to further improve the paper.

---

### Decision · Program_Chairs · 2026-04-30

**Decision:**

Accept (regular)

**Comment:**

This paper studies the online detection of LLM watermarks in sparse, mixed-document streams. The paper proposes  LORD-GOF, a method combining document-level goodness-of-fit (GoF) p-values derived from token-level pivotal statistics with the LORD++ online multiple testing procedure.

Reviewers pointed out the following strengths:
1. The paper addresses a realistic and practically relevant deployment scenario: online detection in sparse, mixed streams rather than offline balanced evaluation.
2. The method is simple, statistically grounded, and easy to understand: GoF-based p-values plus online FDR control via LORD++.
3. The empirical evaluation explores several factors of practical applications and shows strong convincing results.

Reviewers also pointed out the following concerns:
1. Limited Methodological Novelty. The core contribution largely consists of combining existing GoF-based watermark detectors with the LORD++ algorithm, without introducing new statistical principles tailored to watermark structure.
2. Some assumptions do not necessarily hold in practice.
3. Much of the empirical advantage comes from outperforming a naive fixed-threshold baseline, which is not a very strong competitor. Latest and stronger baselines need to be added.